# A Collaborative Adaptation Game for Promoting Climate Action: Minions of Disruptions™

Minja Sillanpää[1], AnaCapri Mauro[1], Minttu Hänninen[1], Sam Illingworth[2], Mo Hamza[3]

[1]Day of Adaptation, Haarlem, 2011 EP, the Netherlands
[2]Department of Learning and Teaching Enhancement, Edinburgh Napier University, Edinburgh, EH11 4BN, Scotland
[3]Division of Risk Management and Societal Safety, Lund University, Lund, 221 00, Sweden

*Correspondence to*: Mo Hamza (mo.hamza@risk.lth.se)

**Abstract.** With the onset of climate change, adaptive action must occur at all scales, including locally, placing increasing responsibility on the public. Effective communication strategies are essential, and adaptation games have shown potential in fostering social learning and bridging the knowledge-action gap. However, few research efforts so far give voice to participants engaging with collaborative games in organisational and community settings. This paper presents a novel approach to studying designer-participant interactions in adaptation games, diverging from traditional learning-focused frameworks. Specifically, it examines Minions of Disruptions™, a collaborative tabletop board game, through the lens of how participant perception aligns with the game's design intentions as described by the game designers and facilitators. Through focus group interviews with designers and facilitators, ten core design intentions were identified and compared with responses from post-game surveys of participants from 2019-2022. Key insights reveal that collaboration and team-building are highly effective frames for climate adaptation. However, some design elements, such as time pressure, can hinder discussion, suggesting a need to balance objectives. The method adopted manages to avoid traditional expert-to-public analysis structures, and places emphasis on the importance of iterative design based on participant insights. This approach provides valuable guidance for future adaptation game designs, demonstrating that games can effectively engage diverse groups and support local adaptation efforts by creating a sense of belonging and collective purpose.

## 1 Introduction

The impacts of climate change are intensifying, manifesting in extreme weather events that are becoming a norm rather than an anomaly (Seneviratne et al., 2021). The increasingly detrimental impacts on people's lives and livelihoods transform climate adaptation from a worst-case scenario to a reality that requires significant investments of resources at all levels: from government-led to individual household-level action (Noll et al., 2022). While adaptation has regionally and sectorally specific hard limits beyond which any adaptive action becomes impossible, concerted action can influence its soft limits, such as through lowering human system-related barriers, including limited financial resources. Today most reported adaptation actions are happening on the individual and household levels (Berrang-Ford et al., 2021) and many adaptation solutions and trade-offs

are best discovered and implemented locally (Moser and Pike, 2015). Therefore, successful society-wide adaptation is currently dependent on increasing local climate awareness (Illingworth and Wake, 2019) and capacity to make informed choices among those who are neither scientists nor policymakers (Whitmarsh et al., 2013).

Prior instances of communicating adaptation to heterogeneous audiences has not resulted in the desired levels of public engagement and commitment (Whitmarsh et al., 2013; Ouariachi et al., 2017). Communication strategies tend to build around an information-deficit model, namely, the assumption that attitude and behaviour change is positively related to an increase in information about a topic; even if the effectiveness of this approach is increasingly questioned in engaging non-scientist audiences (Illingworth and Wake, 2019; Andersson et al., 2019; Badullovich et al., 2020). A so-called knowledge-action gap is used to describe a situation where the audience has the appropriate level of information, yet no adaptive behaviour emerges (Flood et al., 2018). Previous studies have found that a focus on the quantity of information may omit important considerations if unidirectionality renders the audience passive (Illingworth and Wake, 2019; Ouariachi et al., 2017; Parker et al., 2016; Illingworth and Jack, 2018); if jargon forms a barrier to comprehension (Illingworth and Wake, 2019); and if negative frames lead the audience to apathy by triggering feelings of overwhelm and hopelessness (Ouariachi et al., 2017; Moser, 2016). Hence, to bridge the gap, there is a call for more dialogical approaches to address the needs of diverse audiences (Illingworth and Wake, 2019; Illingworth, 2020; Kumpu, 2022).

The attention toward climate adaptation games has increased substantially in the last decade (Flood et al., 2018). There is increasing evidence pointing at the ability of games to address a wider range of audiences (Illingworth and Wake, 2019; Ouariachi et al., 2017; Parker et al., 2016), and enable social learning (Ouariachi et al., 2017; Flood et al., 2018; Den Haan and Van der Voort, 2018; Rumore et al., 2016). The field is still emerging, with several questions remaining unanswered, including how to make the game messages fit for audiences with non-science and non-policy backgrounds (Parker et al., 2016; Galeote et al., 2021; Neset et al., 2020).

This paper brings new insights into this topic by introducing a case study: an analogue and collaborative tabletop game, Minions of Disruptions™ (MoD). The game, developed by a Dutch non-profit organisation Day of Adaptation in 2019, has an explicit objective to engage diverse organisations and communities in collective climate adaptation, regardless of their prior affiliation with climate change. Researchers conducted a focus group exercise with game designers and facilitators to determine the intentions behind the design of MoD and contrasted this information with participants' post-game survey responses, in a new method to study the designer-participant interaction in adaptation games. This method sought to avoid replicating expert-to-public communication structures by including the whole experience, not just participants as objects of study, as a part of the analysis (Illingworth, 2020).

This article addresses the overarching question of what guidelines should be taken into consideration when designing analogue climate adaptation games for the public. It is further explored in three specific sub-questions regarding the intentions behind the game design of MoD according to the designers and game facilitators, the extent to which the design intentions behind MoD are perceived by the game participants, and how the reception of the design intentions by the game participants align with the original objectives of the game.

This article is structured as follows:

- Sect. 2 discusses existing knowledge about adaptation games, and highlights gaps in relation to designing for the general public.
- Sect. 3 outlines the MoD case study and discusses the chosen research approach, data collection, and analysis.
- Sect. 4 introduces the results in two parts: design intentions and their alignment with the participant experience.
- Sect. 5 relates the findings to previous research efforts, suggests a guideline for adaptation communicators, proposes future research directions, and outlines strengths and limitations of the study.
- Sect. 6 offers conclusions and key insights of this method.
- Sect. 7 provides supplemental information.

## 2 Background: climate adaptation games

Generally, climate games can be thought to have three kinds of objectives: (1) increasing awareness of climate challenges; (2) increasing general knowledge, familiarity, and understanding; and (3) encouraging solution-finding and action-taking (Reckien and Eisenack, 2013). Additionally, adaptation games have a broad topical range including resource and environmental management, farming, coastal development, supply chain logistics and transport, disaster preparedness and response, food security, global impacts and change, policy, and climate services (Flood et al., 2018).

Flood et al. (2018) argue that even though the field is emerging, games are proving to be powerful communication tools, helping to realise climate change adaptation faster than with other existing means. They are additionally proposed as a way to address the aforementioned knowledge-action gap (Flood et al., 2018; Ouariachi et al., 2020). Adaptation and climate games succeed in not only creating cognitive, but also normative and relational learning (Flood et al., 2018; Den Haan and Van der Voort, 2018; Rooney-Varga et al., 2020). The reason for their effectiveness is understood to be a consequence of the way games package and deliver information: they are often narrative-based (Flood et al., 2018), more memorable (Parker et al., 2016; Ouariachi et al., 2017), able to capture and explain complexity (Parker et al., 2016; Flood et al., 2018; Den Haan and Van der Voort, 2018), and relatable, as they make use of familiar and locally relevant themes (Parker et al., 2016; Rumore et al., 2016; Galeote et al., 2021; Mitgutsch and Alvarado, 2012; Rodela et al., 2019; Nussbaum et al., 2015). The style of participation is also different because it invites the participants to assume roles and makes information reception more active (Parker et al., 2016; Flood et al., 2018; Galeote et al., 2021; Fjællingsdal and Klöckner, 2020). The participants get the opportunity to explore real-time hypothetical scenarios, which can help make connections between action and impact (Flood et al., 2018; Fjællingsdal and Klöckner, 2020).

From the perspective of local level adaptation, multiplayer collaborative games are a particularly interesting avenue because they provide the possibility for relational learning, which includes gaining a better understanding of others' mindsets and increasing trust and the ability to cooperate (Den Haan and Van der Voort, 2018). Moreover, social simulations can enhance affective learning paths, namely, associating emotions such as concern, importance, and outrage with climate change (Rooney-

Varga et al., 2020). If designed as a dialogical tool, games can help share and co-produce local knowledge (Flood et al., 2018; Den Haan and Van der Voort, 2018) and create an out-of-the-ordinary space for conversation (Flood et al., 2018; Rumore et al., 2016; Fjællingsdal and Klöckner, 2020) with fewer knowledge hierarchies (Illingworth and Wake, 2019; Illingworth, 2020; Rodela et al., 2019). Enabling such conversations is key in increasing normative reflexivity at the group level, which could change or facilitate internal decision-making (Flood et al., 2018; Rumore et al., 2016; Rodela et al., 2019). Games have also been seen to increase the perceived importance of cooperation, empathy, and respect toward other perspectives (Rumore et al., 2016; Galeote et al., 2021; Rodela et al., 2019; Abspoel et al., 2021), augment feelings of trust and ownership (Flood et al., 2018; Ouariachi et al., 2020), and even solve conflicts (Medema et al., 2016). Additionally, they may increase optimism about the effectiveness of local cooperation (Rumore et al., 2016; Galeote et al., 2021; Ouariachi et al., 2020).

While there is much traction around games, research gaps remain. Few climate games known to research propose collective-level solutions, create dialogue, focus on affective learning, or aim at achieving direct impact (Gerber et al., 2021). On the other hand, games enhancing cognitive learning are the highest represented in research, whereas normative and relational learning are rarely addressed (Den Haan and Van der Voort, 2018). Furthermore, games can fail to reach the objectives set for them: they sometimes narrate roles that the participants do not identify with (Galeote et al., 2021), fail to form linkages with real-life (Fjællingsdal and Klöckner, 2020), are not relevant (Lankford and Craven, 2020), or overwhelm participants with information, curtailing dialogue (Illingworth, 2020). There is an additional degree of ambiguity about the optimal medium: some studies question the effectiveness of digital games (Boomsma et al., 2018), whereas others find that, for example, video games deliver best results (Olivares-Rodríguez et al., 2022).

There are different climate game designs to address diverse target audiences, such as students, policymakers, professionals, or the general public (Gerber et al., 2021). The "general public" in particular is often loosely defined, but here it is understood as a group that engages little with climate change in their day-to-day; they do not have a science background, nor do they work with the topic professionally. This group tends to be the least represented in climate game reviews (Parker et al., 2016; Galeote et al., 2021; Neset et al., 2020), and generally in science engagement strategies (Illingworth and Jack, 2018). Gaining a better understanding of this interaction can help explain why the participants cannot always relate to the game content, or what kind of information might overwhelm them. The public may have an attitude, cognitive style, or mode of learning that diverges significantly from that of the communicators and of each other, and therefore presents a particularly important dimension of study. Exploring this topic might, therefore, give answers as to what contributes to gaps between knowledge and action, and how they could be bridged.

Effective climate communication requires that the audience(s) is determined and well-known in advance (Illingworth and Wake, 2019) and that their needs are understood (Ouariachi et al., 2017; Flood et al., 2018; Monroe et al., 2019). Therefore, it is proposed that this study enhances the game field through deepening the understanding about the needs of the audience and capturing their interaction with the game and the communicators. Designers play a key role in the outcome of the game, as they ultimately decide what information gets communicated via the game and in what way, thereby dictating what success looks like (Fjællingsdal and Klöckner, 2020). Scientific articles on climate games tend to focus on measuring the participant

experience pre-, post-, and post-post game events (Flood et al., 2018; Den Haan and Van der Voort, 2018) and by doing so
somewhat omit this relationship. In the interest of understanding how games could help realise rapid local-level adaptation,
design and engagement guidelines are needed to inform future designs and game iterations.

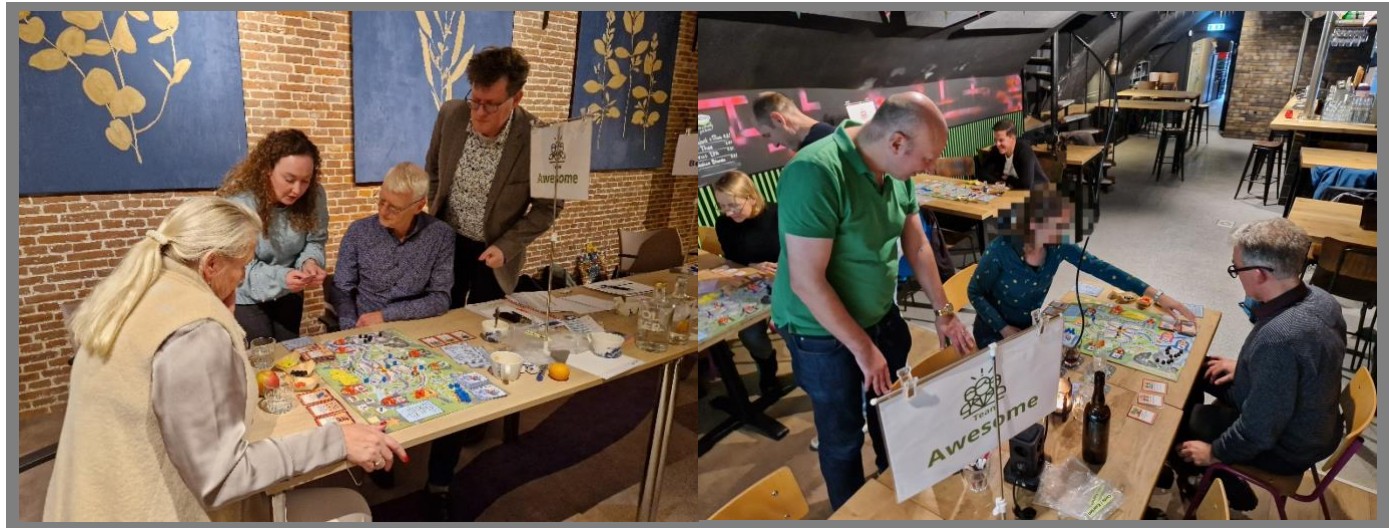

**Figure 1. Game participants playing Minions of Disruptions™. © Day of Adaptation 2023.**

**3 Method**

**3.1 Minions of Disruptions™**

This research paper studies a collaborative and analogue tabletop game, Minions of Disruptions™ (MoD), created in 2019 by
a Dutch non-profit organisation, Day of Adaptation (https://dayad.org/). The organisation explores and innovates on climate
communication, targeting specifically groups that tend to be left out of the conversation. "Game Day," a facilitated gameplay
experience, is one of its communication tools. The game can be played by anyone, as there is no strictly defined target audience.
However, there is a general player typology: players are predominantly adults of various ages or university students,
representatives of the same or somehow affiliated communities and organisations, and most of the participants are not climate
professionals nor students of climate sciences. All groups enjoy the privilege of time to dedicate for such an activity, the costs
of which are covered by their employer or administration.
The data used in this study were collected by Day of Adaptation for monitoring and evaluation purposes (see Table 1 for an
overview). There are both online and in-person versions of the same game activity with an even split between events organised
in the Netherlands versus other countries. The range of organisation type is broad, and while the survey did not systematically
measure the general level of climate knowledge or the level of gaming experience of the participants, anecdotally it can be
said that it varies both between events and within groups. For instance, sometimes a Game Day might be organised by an
employee who is part of a sustainability committee at the workplace. This individual is bound to have a different level of

background knowledge in comparison with their colleagues. An average player is aware of the basics of climate change, however, not necessarily familiar with its causes and consequences. Some groups or individuals might be taking some collective climate action already, whilst others are only getting started, and hope to use the event to kickstart and get their team or organisation engaged and involved.

**Table 1. The dataset used in this study, comprising 18 Game Days that took place between 2019 and 2022.**

| ID | Date (y-m-d) | Organisation type | Country | Game Version | Participants | Surveyed Participants | Survey Participation (% of Participants) | Sample Distribution (% of total surveyed) |
|----|----|----|----|----|----|----|----|----|
| 1 | 2019-12-02 | University | Netherlands | In-person | 25 | 19 | 76 | 13.57 |
| 2 | 2020-04-16 | Activist Group | Netherlands | Online | 3 | 2 | 66.7 | 1.43 |
| 3 | 2020-06-28 | Association | Netherlands | In-person | 5 | 4 | 80 | 2.86 |
| 4 | 2020-08-19 | Bank | Netherlands | In-person | 12 | 2 | 16.7 | 1.43 |
| 5 | 2021-01-24 | Community of Climate Professionals | Netherlands | Online | 60 | 14 | 23.3 | 10.00 |
| 6 | 2021-04-05 | Activist Group | Chile | Online | 4 | 3 | 75 | 2.14 |
| 7 | 2021-04-23 | Non-profit Organisation | Germany | Online | 9 | 6 | 66.7 | 4.29 |
| 8 | 2021-04-26 | University | Philippines | Online | 20 | 20 | 100 | 14.29 |
| 9 | 2021-04-28 | Social Movement | UK | Online | 8 | 5 | 62.5 | 3.57 |
| 10 | 2021-05-06 | Non-governmental Organisation | Netherlands | Online | 7 | 1 | 14.3 | 0.71 |
| 11 | 2021-05-12 | University | Mexico | Online | 13 | 10 | 76.9 | 7.14 |
| 12 | 2021-09-03 | University | Netherlands | In-person | 33 | 1 | 3.0 | 0.71 |
| 13 | 2021-09-03 | Cross-regional government mandated body | Netherlands | In-person | 19 | 16 | 84.2 | 11.43 |
| 14 | 2021-10-01 | University | Netherlands | Online | 35 | 1 | 2.9 | 0.71 |
| 15 | 2021-10-30 | Development Institution | Saint Vincent | Online | 9 | 6 | 66.7 | 4.29 |
| 16 | 2021-12-08 | University | Sweden | In-person | 25 | 10 | 40.0 | 7.14 |
| 17 | 2022-05-24 | Private Company | Australia | Online | 10 | 5 | 50.0 | 3.57 |
| 18 | 2022-05-25 | Private Company | Australia | Online | 24 | 15 | 62.5 | 10.71 |
| | **Total** | | | | **321** | **140** | | **≅100** |

### 3.1.1 The gameplay

The standard format for a session is a three-hour game activity, which can take place either in person or online. In-person events use physical versions of the game, while online events utilise an online conferencing software and Tabletopia. Tabletopia is a digital sandbox system for playing board games with no AI to enforce the rules, which allows for the game pieces to be manipulated by the players as they please, creating a life-like board game situation. Because the online version provides no feedback or automation, the in-person and online experiences are comparable for the purposes of this study. Groups opt to play either a community or organisation version of MoD (see Fig. 1 for an example board). While the basic rules of the game are the same regardless of the version, the content is somewhat adjusted: the community version focuses on services such as housing, and the organisation version on operational functions. Sometimes the game content is even further adjusted, if requested by the community/organisation during the planning phase.

All events begin with splitting the group into teams of 3-4 people, each with their own board. The teams are given the basic rules of the game after which they learn the game experientially. All teams have the same goal: to implement climate actions strategically and collaboratively in a game world where increasing carbon levels in the atmosphere increasingly slow them down and inflict continuous disruptions. Players move around the board pathways trying to remove tokens that represent vulnerability to the different sectors, while trying to protect and increase the resilience of these critical services to future climate disruption. The tokens signifying disruptions are removed by using the team's mutual funds for climate action (both mitigation and adaptation related), however the team needs to act fast because the disruptions increase exponentially. Different cards drawn by each player during their turn and sound effects played by the facilitator can alter the gameplay in either helpful or hindering ways. The team also needs to balance financial costs and can negotiate with other teams to move forward faster. Occasionally they are invited to share real-life knowledge and experiences, which have an impact on their gameplay. A team wins the game by protecting five of their organisation/community's essential sectors against disruptions, indicating climate resilience.

Gameplay takes 60-90 minutes, with the remaining time used for a brief warm-up and facilitated debrief. Depending on the participants' wishes the facilitators may include supporting team-building activities, and introduction of basic terminology (e.g., mitigation, adaptation). The debrief is structured into three parts. The first part focuses on a review of the game experience, including discussions of how realistic the game felt and how the teams interacted. The second part connects the game play to reality, including what climate change looks like for the organisation/community in question. The third part brings the discussion home to climate action and allows participants to discuss how they will take the Game Day experience back into their lives, including the barriers to action they may encounter and how to mitigate these real-life disruptions. This structure aligns more closely with the view of generating knowledge through action rather than trying to impart knowledge first and then expecting participants to transform this into action via the game (Crookall and Thorngate, 2009). The goal being that action in the game translates to knowledge and learning, and then into real-life action.

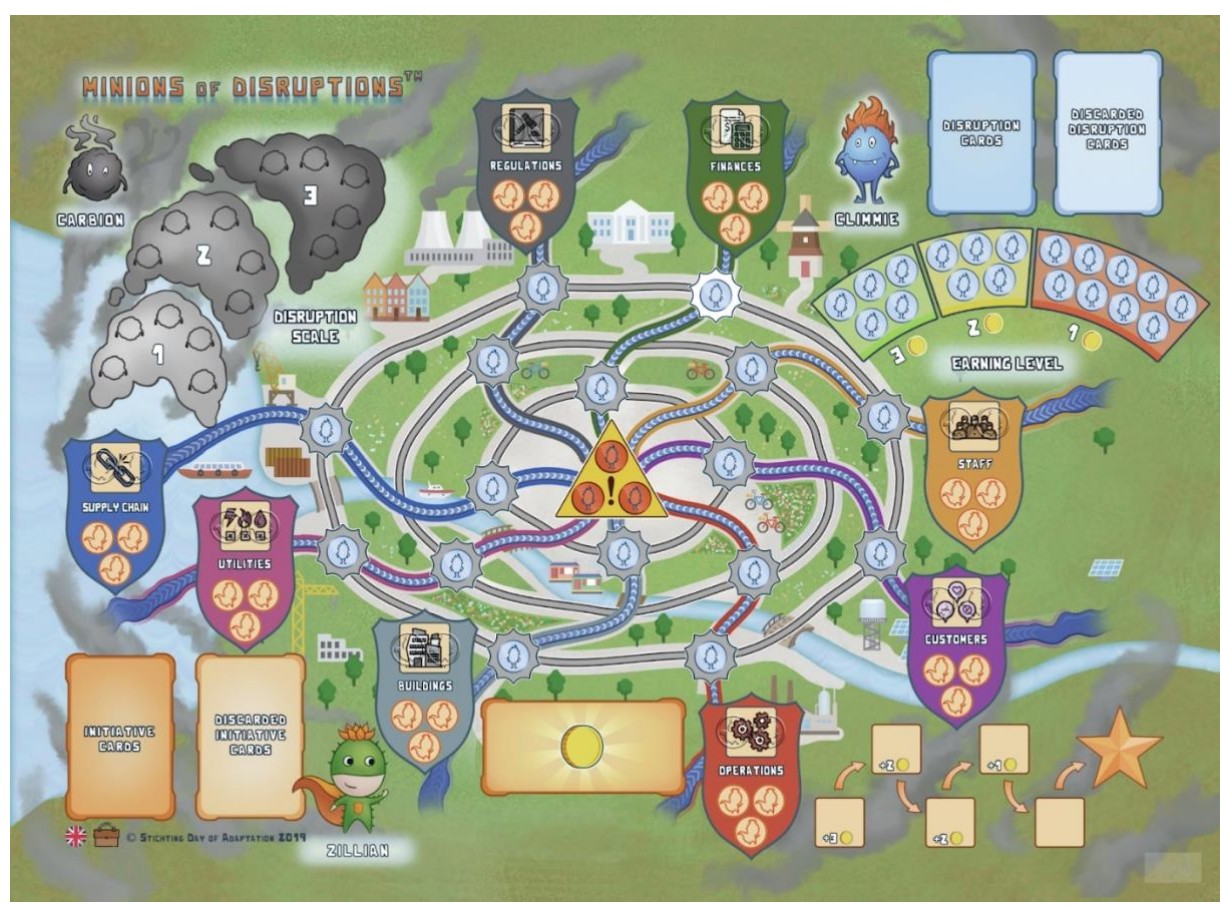

**Figure 2. The visual layout of the Minions of Disruptions™ game board, which models climate disruptions in an organisation. The operational functions, or "shields", include operations, customers, staff, finances, regulations, supply chain, utilities, and buildings. © Day of Adaptation, 2019.**

### 3.2 Methods and datasets

This paper adopts a novel approach that combines data from game designers and facilitators with data collected from game participants. Unlike some participant experience focused methods, which commonly evaluate games by observing gameplay or analysing participant surveys only (Flood et al., 2018; Den Haan and Van der Voort, 2018), the purpose of this method is to assess games as transitional objects, which may or may not succeed in conveying what the designers and facilitators of the game intended. In other words, this method forms a connection between design intent and how the gaming experience is perceived by participants by not only asking how the participants behaved, and what they perceived, but also what the original intention of the designers and facilitators was.

The reason for adopting such an approach over the more common participant observation is to address what has been found by others previously, namely, that intention-based designs should be analysed and understood in relation to their purpose

(Neset et al., 2020). While this remains true, there are important factors that get omitted if it is taken for granted that the

designed purpose is fixed and unaffected by those who play the game. As previously found, there are confounding factors that

mislead findings when measuring for social learning from games, for example, preheld notions of the game or gaming in

general, the agency of the facilitators, and prior in-group relations (Den Haan and Van der Voort, 2018). In actuality, the

participants construct their own experience, which may or may not stand in congruence with the intentions of the designers.

Therefore, a game design may lead to emergent qualities. This method aims to capture such qualities, which may be

unknowingly omitted when focusing on participant experience only. By first addressing a designer perspective followed by a

participant perspective, a journey from a design intention to a lived participant experience is constructed, which allows one to

study the contrasts between the two. For the purposes of this study, this approach remains qualitative due to the subjective and

narrative nature of the data and the lack of strict uniformity of the game events. The conclusions drawn through this approach

contribute to a validated foundation off which future quantitative studies could be built.

### 3.2.1 The designer perspective

A 1.5-hour online focus group interview was organised in April 2022 with three game designers/facilitators and two facilitators

from Day of Adaptation. Eight participants in total were invited to take part, but three were unable to attend. This sample

represents the majority of the designers, and at the time of the study, approximately a third of the active facilitators. The

researchers set up the focus group with the objective of capturing design intentions, meaning, what kind of messages the

designers and facilitators wanted to convey to the audience and what kind of elements they designed to fulfil this objective

(e.g., tangible game pieces, rules, etc.). The participants were informed about the purpose of the focus group prior to and during

the data collection, and they all consented to being featured in this research.

The session was managed with Zoom and Miroboard-platforms. As a warm-up, the participants took turns listing what different

game elements they could remember, adding to each other's knowledge. In the second part, these game elements were

momentarily set aside, and the participants were asked to reflect on high-level design intentions of the game and what core

ideas it aims to address. In the third part, the game elements were reintroduced and the participants were asked to connect and

cluster them with the design intentions.

Focus group as a method of data collection is often used when interviewees have a history of working together, when it is

assumed that benefits can arise from immediate cross-checking of statements on a group-level, and when researchers wish to

generate representative data whilst being mindful of participants' and their own time constraints (Creswell, 2013). In this case,

most focus group participants and all designers had worked together previously. Given that three years had passed since the

creation of the game, and two of the participants have not been involved with Day of Adaptation since, the focus group was

intended to serve as a way to have an agreeable re-encounter, to help refresh memories, and bring about consensus-based

answers to the interview questions.

This method has its pros and its cons. For the pros, it poses less pressure on a single participant and, therefore, given

participants' busy schedules, it was considered the best option. Additionally, the organisers aimed to make the experience as

stress-free as possible so, in addition to the researcher in charge of leading and directing discussion, two co-organisers joined
the session to manage the technical side and to note observations. No technical difficulties emerged, however, in the case they
would have, the session would have been temporarily paused or postponed to ensure good quality discussion. The participants
could, thus, simply focus on thinking, commenting, and answering questions, which helped to make the best use of their time
and generate a great quantity of data in a short amount of time. Another benefit of the method was that there was no need to
cross-check answers as that could be done in real time during the focus group.

For the cons, a focus group, as any group situation, is bound to follow pre-established group logics and power dynamics, which
may influence which data are generated or excluded by the group. Moreover, such a form of interaction may not suit all
personality types and can favour individuals who are more inclined to speak in a group setting. Further, with small group sizes
and self-reporting, there is the potential for biases such as social desirability bias, in addition to memory recall errors and
reliance on subjective interpretations of individual experiences. In order to mitigate issues related to memory, the participants
first got time to inspect the game board to trigger their visual recollection. The researchers aimed to enable such a space through
specific design choices: in most cases participants were asked to answer in randomised turns, instead of giving an open floor,
and they were also directly asked to comment on each other's contributions. Furthermore, both the designers and game
facilitators were included in the same session. This allowed the game facilitators to pose questions to the designers, which
could help challenge the internal dynamic of the designer group.

**3.2.2 The participant perspective**

The audience perspective is taken from a standardised post-game survey that all game participants were asked to fill out at the
end of their group's Game Day (see Appendix A for a list of the survey questions). This survey is designed to collect monitoring
and evaluation data for Day of Adaptation and was not originally intended to be used for research as such. The organisation
gave consent to analysing these data, and the researchers received it anonymised so that only the organisation names and some
basic demographic data were retrievable. The survey participants have not given their explicit consent for this research, but
their participation in the original post-game survey was voluntary and they could opt-out from any question. To protect the
integrity of the participants, demographic data are only treated on a general level so that it cannot be connected to any
organisation or individuals. The age of participants spans from 18 to 65+, with an average age of 32 years. More than 60
262  percent of the participants identify as female, 36 percent as male, and 2 percent as non-binary. The participants represent a
wide variety of organisations (see Table 1 for the breakdown of organisations included in the analysis). Anecdotally it can be
said that apart from the student groups, the groups are teams that work together directly or under the same organisation,
representative of a variety of job levels.

Previous survey research on games has found that not only is it a quick and inexpensive method to measure immediate impact,
but it can also be considered robust insofar as the data are representative of a great number of game events (Flood et al., 2018).
In total there are 140 survey answers from 18 game activities, played between 2019-2022, including both the online and in-
person versions of the game. The survey consists of multiple choice and open field questions, but only the latter was included

in this study, as it was considered better suited to answer the research questions of this paper. This means that no connection
is drawn between sample demographics and the answers, but the focus is on the general participant level. Comparing and
contrasting between types of groups and institutions would add depth to our understanding of tailored climate communication.
This is excluded from the scope of this research, however, given that the researchers deal with third-party data in the selection
of which they had no part to play, nor did they receive sufficient background information on the profiles of the participants. It
was, therefore, deemed that generalisations on groups would be untenable.

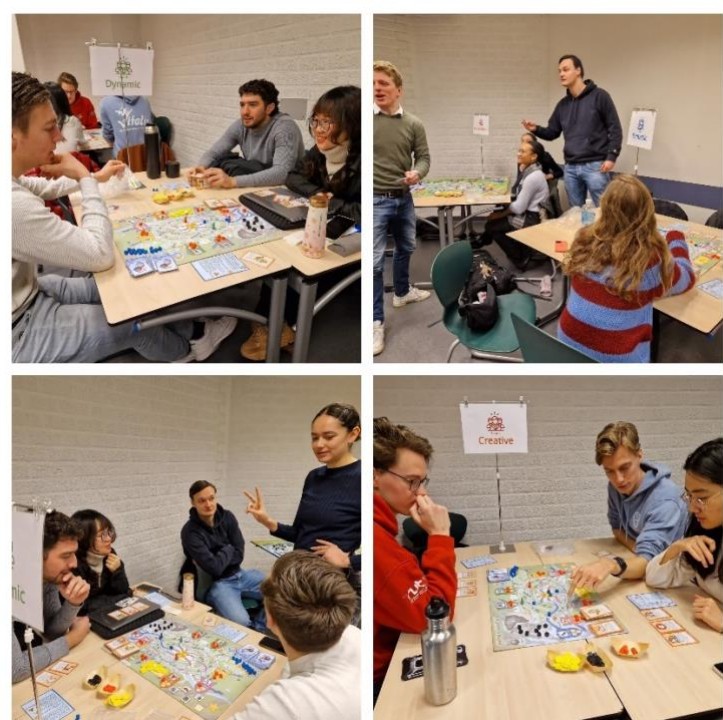

**Figure 3. Game participants playing Minions of Disruptions™. Images illustrate both within team decision-making as well as negotiations between the different teams. © Day of Adaptation, 2024.**

### 3.3 The analysis

The analysis consisted of two steps. In the first step, the data collected during the focus group inquiry were processed; the recording was transcribed, and participants were anonymised. During the focus group, the participants agreed in consensus upon ten design objectives and related them to game design elements. While engaging in dialogue, their answers were simultaneously modelled on a Miroboard by the organisers. The participants could immediately react to the accuracy of the visual representation via screen-sharing. To ensure that all of the expressed ideas were correctly interpreted after the focus

group, the transcription and the language used by the participants was contrasted with the visual representation. The transcription was prioritised in order to capture ideas that might have been omitted during the interpretation process.

The second step of the analysis mapped out how game participants perceived the game as a transitional object conveying the ten design intentions. Once the ten design intentions were established, two researchers conducted independent Excel analyses that coded the open-field questions of the post-game survey for all participants, both into the design intention categories and then for positive (1), negative (-1), or neutral (0) alignment with the design intentions. These scores were then averaged to determine an "alignment score" for each design intention. Statements were permitted to have no more than two design intention categorisations as an analytical boundary imposed by the researchers. It is recognised that this may lead to a simplified version of reality.

The aim was to connect entries with evidence for and against the fulfilment of a design objective. The two independent analyses were compared and negotiated between the researchers to arrive at a mutually agreed upon categorisation. This information is discussed both for the whole sample as well as divided based on how the game was presented, either online or in-person, to demonstrate the general reception of the game as well as to observe any potential variance based on experience. Individual groups were not analysed on their own due to wide variation in the number of respondents per session. While this approach could potentially lead to one group's poor experience skewing the analysis, it was determined to be acceptable because of the consistency observed in the data between groups.

## 4 Results

### 4.1 The design intent

The focus group participants elaborated on ten design intentions that they aimed to achieve with the game, as well as various design elements included to achieve the intentions. The design elements have been categorised in line with an applied framework combining typologies from Gerber et al. (Gerber et al., 2021), Lankford and Craven (Lankford and Craven, 2020) and Razali et al. (Razali et al., 2022) and are elaborated upon in Appendix B. The following ten design intentions, in alphabetical order, were agreed upon by the focus group participants:

1. **Adaptive Action:** Addressing climate action both from mitigative and adaptive perspectives.
2. **Climate Science:** Increasing awareness of basic climate change elements in daily lives, as well as the anthropogenic cause-and-effect of climate change.
3. **Collaboration:** Addressing both individual and collective action but taking the organisation/community as the starting point.
4. **Language:** Communicating with simple language so that the game is accessible for a wider audience with varying education levels and interest.
5. **Moderation:** Autonomous gameplay with minimal moderation to emphasise the agency of the team.

6. **Organisational Relations:** Increasing understanding of the complexity of connectivity and interaction of essential services and functions of organisations and communities in an era of climate change.

7. **Psychological Resilience:** Triggering reflections within participants on adjusting to a new climate and its consequences.

8. **Relatability:** Being relatable through incorporating relevant current events, research, and unique examples from participants' lives.

9. **Setting:** Creating a fun and welcoming space to inspire and increase motivation to act through a positive solution-frame.

10. **Team-building:** Increasing intra-organisational conversations despite existing hierarchies; learning to collaborate and enhancing team-building to build bridges and synergies that can help with action-taking.

**4.2 The participant experience**

The ten game design intentions identified by the focus group participants created a framework through which to measure the impact of the game. All open-field responses of the post-game survey were coded into these intention categories. One hundred and forty participants responded to the survey, with 52 respondents from in-person Game Day events and 88 from online events. Not all participants answered every question, and 115 statements were omitted from the analysis due to ambiguity. Sixty-nine statements fell into two different design intention categories and were therefore counted twice. In total, 265 unique responses were included in this analysis, combined with the 69 responses falling into two categories for a total of 334 statements to be categorised (89 in-person and 244 online). Raw participant and statement numbers can be found in Appendix C.

All design intentions were represented in the survey responses, though with varying frequency. *Adaptive Action* was the most represented design intention (20.96% of total), while *Psychological Resilience* was the least represented as a percentage of the total responses (1.5%) (Fig. 2). Following *Adaptive Action* were *Setting* (15.27%), *Moderation* (14.07%), *Collaboration* (13.77%), *Climate Science* (11.98%), *Relatability* (7.19%), *Language* (6.29%), *Organisational Relations* (5.09%), and *Team-building* (3.89%).

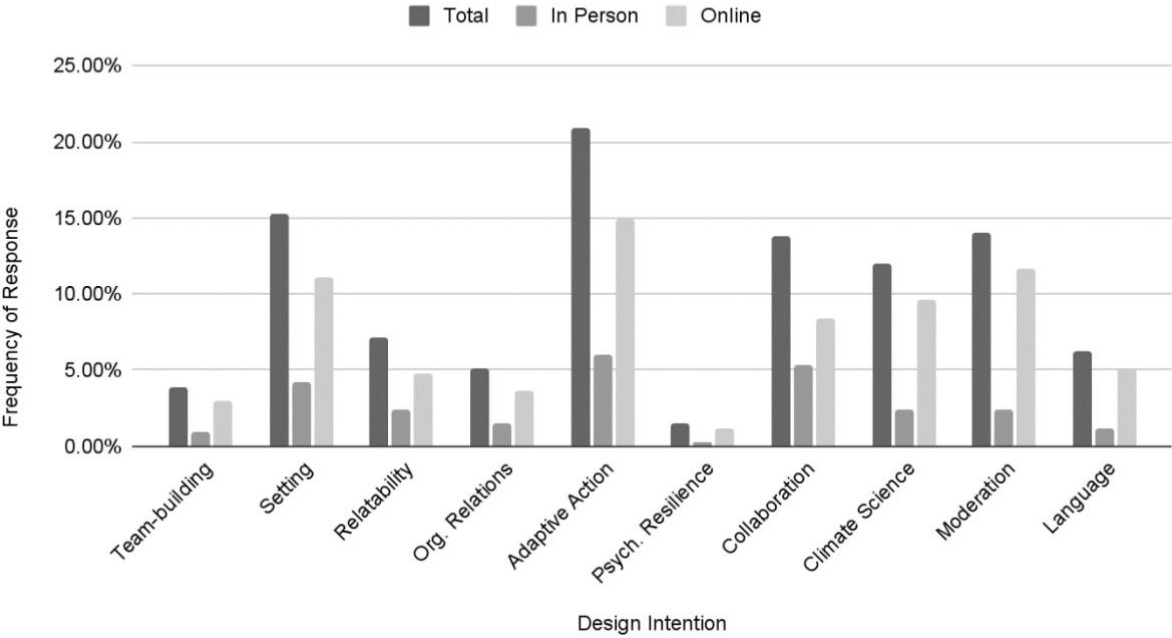

**Figure 4. Percentage of responses (% of total) categorised by design intention for in-person and online events and the total for each design intention.**

### 4.2.1 In-person versus online events

In-person participants accounted for 37% of survey respondents and approximately 26% of statements analysed. All design intentions were represented in responses as shown in Figure 4.

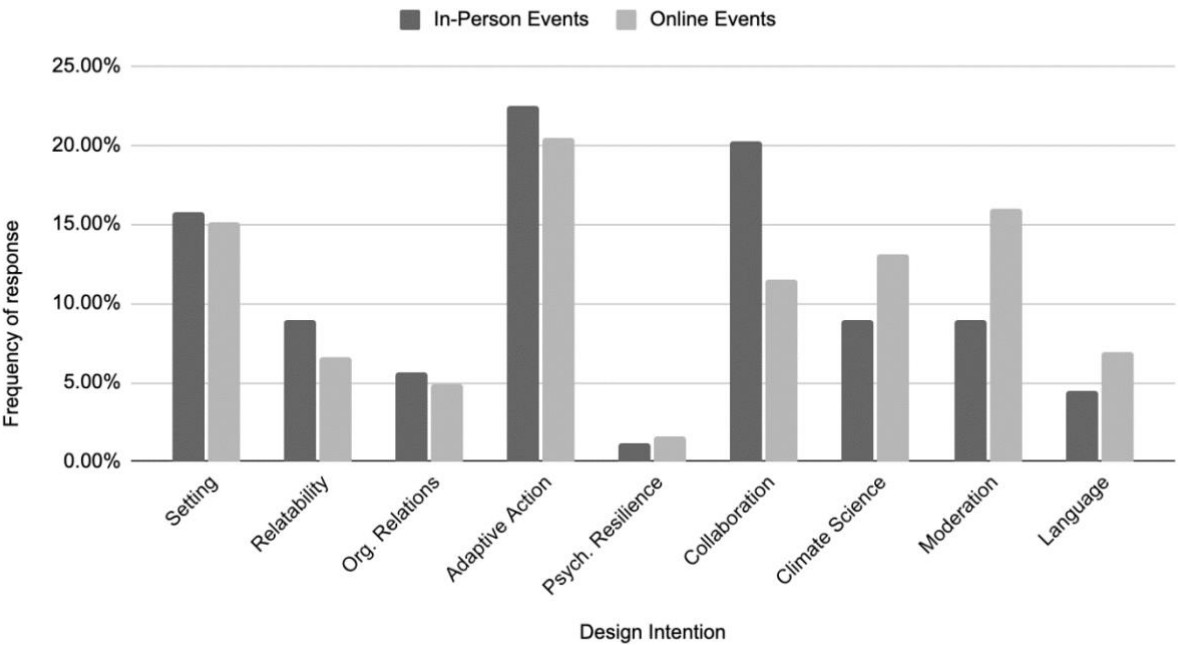

## Event Statement Breakdown (% in each event type)

**In-Person Events** ■ **Online Events** ■

*Frequency of response* (y-axis: 0.00%, 5.00%, 10.00%, 15.00%, 20.00%, 25.00%)

Design Intention (x-axis: Setting, Relatability, Org. Relations, Adaptive Action, Psych. Resilience, Collaboration, Climate Science, Moderation, Language)

**Figure 5. Distribution of design intention occurrence between in-person events and online events (percents from within each event**
**type).**

Though all intentions were mentioned, 42.7% of all statements fell into just two categories: *Adaptive Action* (22.5%) and
*Collaboration* (20.2%). *Setting* (15.7%), *Climate Science* (9.0%), *Relatability* (9.0%), and *Moderation* (9.0%) also had a
combined total of 42.7%, with these six design intentions dominating 85.4% of the statements included. The remaining four
intentions*, Organisational Relations*, *Language*, *Team-building*, and *Psychological Resilience*, were the least represented.

Participants in online events accounted for approximately 63% of survey respondents and 73% of statements analysed. All
design intentions were represented in responses as shown in Fig. 3, with a slightly more balanced distribution than noted in
the in-person survey responses.

For online events, *Adaptive Action* was the most referenced intention at 20.4%, which is similar to the frequency found in in-
person events (22.5%). *Moderation* and *Setting* were nearly tied for the second-most referenced design intention (15.9% and
15.1%, respectively), followed by *Climate Science* (13.1%), and *Collaboration* (11.4%), for a combined total of 75.9% of
statements analysed. The remaining five design intentions, Accessible *Language*, *Relatability*, *Organisational Relations*,
*Team-building*, and *Psychological Resilience* accounted for the final 24%. Except for *Relatability*, the least represented design
intentions are consistent between in-person and online respondents.

### 4.2.1 Design intention and response alignment

While the initial part of this analysis demonstrates the frequency of the design intentions in survey responses, additional analysis was required to determine whether the statements align with or contradict the game designers' original intentions. Of the ten design intentions, all except *Language* and *Moderation* had overall positive averages in the survey responses (-0.33 and -0.38, respectively). *Team-building* and *Collaboration* had the highest overall averages at 1.00, followed closely by *Organisational Relations* (0.94) and *Climate Science* (0.90). *Adaptive Action* (0.80), *Relatability* (0.75), *Psychological Resilience* (0.50), and *Setting* (0.35) complete the list of positively aligned survey responses (See Table 2).

**Table 2. Alignment score for each design intention, including overall average and adjustments for in person and online events. Higher averages indicate closer alignment.**

| Design Intention | Overall Average | In Person Average | Online Average |
|---|---|---|---|
| Adaptive Action | 0.80 | 0.60 | 0.88 |
| Climate Science | 0.90 | 0.75 | 0.94 |
| Collaboration | 1.00 | 1.00 | 1.00 |
| Language | -0.33 | -0.50 | -0.29 |
| Moderation | -0.38 | -0.50 | -0.36 |
| Organisational Relations | 0.94 | 1.00 | 0.92 |
| Psychological Resilience | 0.50 | 1.00 | 0.33 |
| Relatability | 0.75 | 0.63 | 0.81 |
| Setting | 0.35 | 0.79 | 0.19 |
| Team-building | 1.00 | 1.00 | 1.00 |

The alignment changes when adjusting for in-person versus online Game Days. For in-person events, *Team-building* and *Collaboration* were joined by *Psychological Resilience*, and *Organisational Relations* at the 1.00 average, while *Moderation* and *Language* remained negatively ranked. The online Game Days maintained the same rankings as the overall average for all intentions except *Organisational Relations* and *Climate Science*.

When comparing the reception between in-person and online events, in-person events had five design intentions scoring lower than the online average (*Moderation*, *Language*, *Relatability*, *Adaptive Action*, *Climate Science*), while *Setting*, *Psychological Resilience*, and *Organisational Relations* scored lower for online Game Days. *Collaboration* and *Team-building* maintained a

1.00                    average    for    both    online    and    in-person    events    (Fig.    4).

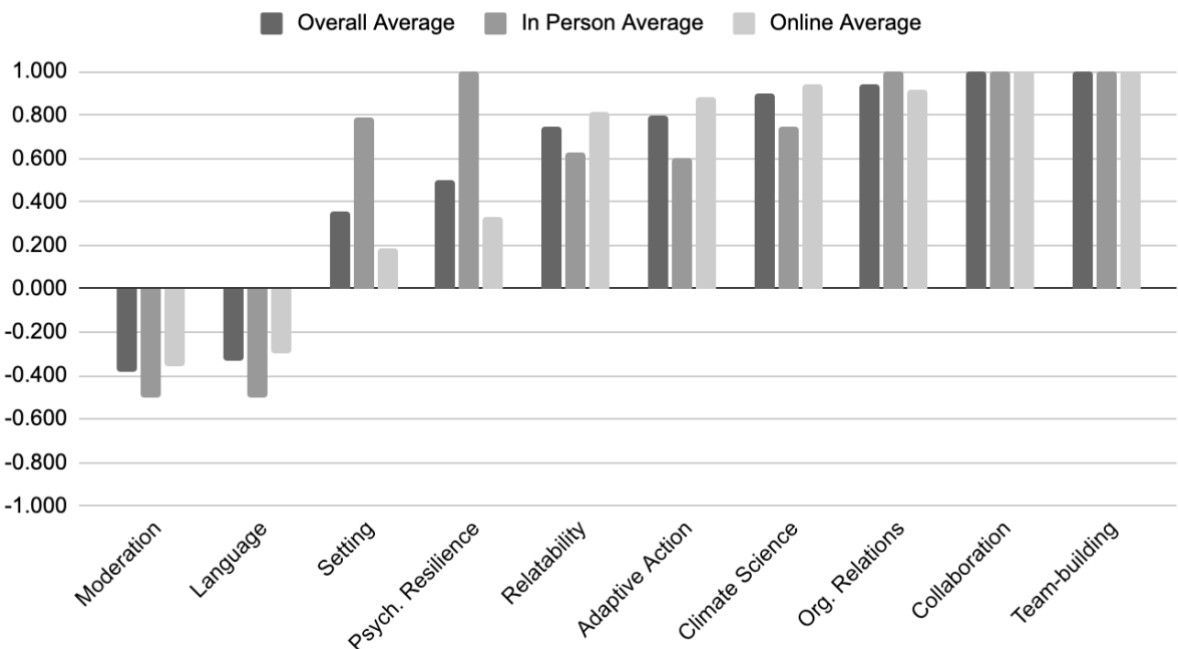

**Figure 6. Alignment scores for all statements to each design intention for overall, in-person, and online Game Day events.**

**5 Discussion**

**5.1 Understanding the results**

**5.1.1 Designer perspective**

The inquiry yielded 10 distinct design intentions and 15 design elements, the latter of which includes aspects of medium,
challenge, reward, level of abstraction, and player interaction, which the interviewees said were incorporated to realise the
design intentions. For conceptual clarity the 10 design intentions are separated here into two categories. The first category is
Primary Objectives, which describes the substantial content of the game. It was found deductively by contrasting the design
intentions with Reckien and Eisenack's (Reckien and Eisenack, 2013) three-fold objectives, and seeing that some design
intentions aim to raise awareness (*Climate Science* and *Psychological Resilience*), increase knowledge, understanding and
familiarity (*Organisational Relations*); and promote action-taking or solution-finding (*Adaptive Action* and *Collaboration*).

The corresponding design elements are shown in Table 3, and a detailed explanation of the connections can be found in
Appendix B.

**Table 3. Presentation of the design intentions and elements of MoD in connection with game objectives as theorised by Reckien and**
**Eisenack.**

| Primary Objective | Design Intention | Design Elements |
|---|---|---|
| Raise Awareness | 1. Climate Science<br>2. Psychological Resilience | Aesthetic Experience<br>Audiovisual Cues<br>Challenge: Time Constraints<br>Challenge: Uncontrollable Events<br>Discussion<br>Medium: Board<br>Medium: Cards for Action<br>Player Interaction: Collaboration/Competition between Teams |
| Increase Knowledge, Understanding, Familiarity | 1. Organisational Relations | Abstraction Level: Qualitative Description<br>Audiovisual Cues<br>Challenge: Limited Funds<br>Challenge: Time Constraints<br>Challenge: Uncontrollable Events<br>Discussion<br>Medium: Board<br>Player Interaction: Collaboration/Competition between Teams<br>Reward<br>Role Play: Explicit Role Assignment with Optional Roleplay<br>Tactical Decision Simulation |
| Promote Action-taking and Solution-finding | 1. Adaptive Action<br>2. Collaboration | Challenge: Uncontrollable Events<br>Discussion<br>Medium: Cards for Action<br>Player Interaction: Collaboration/Competition between Teams<br>Player Interaction: Team Collaboration<br>Reward<br>Tactical Decision Simulation |

The remaining five design intentions, *Language, Moderation*, *Relatability*, *Setting* and *Team-building*, relate less to the game's
content, but rather prescribe how the substance is to be conveyed. It was found that they closely correspond to the general
climate change engagement framework by Ouariachi et al. (Ouariachi et al., 2020), as illustrated in Table 5. Here they are
referred to as Secondary Objectives, as they are not lone standing, but support reaching the Primary Objectives. For instance,
what the engagement framework defines as 'Concrete' is well-aligned with what the designers call *Language*: both aim to

package information in a way that is accessible and relevant to the audience in question who is expected to respond better to less abstract information.

**Table 4. MoD's design intentions and elements connected with Ouariachi et al. climate engagement framework. The design intentions were connected to an objective in the framework with most resemblance in terms of purpose.**

| Secondary Objective | Design Intention | Design Elements |
|---|---|---|
| Achievable, Credible and Identity-driven | Relatability | Abstraction Level: Qualitative Description<br>Audiovisual Cues<br>Challenge: Uncontrollable Events<br>Discussion<br>Medium: Board |
| Concrete | Language | Aesthetic Experience<br>Kinaesthetic Experience<br>Character Design<br>Discussion |
| Social and Reward-driven | Team-building | Discussion<br>Moderation Type: Instructionist with constructionist elements<br>Player Interaction: Collaboration/Competition between Teams<br>Player Interaction: Team Collaboration<br>Reward<br>Role Play: Explicit Role Assignment without role play<br>Tactical Decision Simulation |
| Fun, Meaningful and Reward-driven | Setting | Audiovisual Cues<br>Challenge: Time Constraints<br>Discussion<br>Moderation Type: Instructionist with constructionist elements<br>Player Interaction: Collaboration/Competition between Teams<br>Player Interaction: Team Collaboration<br>Reward |
| Experiential Learning | Moderation | Discussion<br>Moderation Type: Instructionist with constructionist elements<br>Player Interaction: Team Collaboration |

Unpacking the game design of MoD confirms the preconceived notion that adaptation games offer the possibility for complex communication. The messages that the designers want to convey are nuanced and specific, but they can be seen

connected to Reckien and Eisenack's higher resolution three-fold division. On the other hand, connecting the specific design intentions with the design elements in Table 3 gives an idea of how the messages are constructed with the help of different game mechanics.

Table 4 shows a blueprint of the engagement strategy that was designed with the intention that it would fit the needs of the general public. By separating design intentions into objectives and engagement strategy, the topic could be separated from the means. The characteristics and needs of an audience need to be understood if they are to be successfully engaged (Flood et al., 2018; Ouariachi et al., 2017, 2020). For future game iterations and without compromising the action messages that the game is aiming to convey, the information gained about the audience through this study can be used to enhance the engagement strategy, specifically focusing on the Secondary Objectives.

**5.1.2 Participant perspective**

Games aiming to achieve social learning can be conceptualised as transitional objects (Den Haan and Van der Voort, 2018). This implies that they are intended as communication vessels that transmit messages and achieve objectives predetermined by designers and facilitators. However, as with any communication, messages about climate change are transformed by the receiver; they do not simply flow unchanged from a designer to the audience (Illingworth, 2020). It, therefore, helps if the audience(s) is determined and well-known in advance (Illingworth and Wake, 2019). This study explored a new way of understanding the participant perspective by contrasting the designers' intentions with a post-game monitoring and evaluation dataset. As the questionnaire was not designed to capture alignment with the design intentions, it can be said with somewhat high confidence that the results organically represent the strongest and weakest communication aspects of the game across the data sample.

Surprisingly, even when controlling for online/in-person interactions, all of the design intentions were referred to by the survey participants. This is interpreted as validating the focus group method used to retrieve the design intentions. Furthermore, it shows that despite the degree of design complexity, the game succeeds in transmitting all of its communication components. Thus, the interesting question becomes where it was least and most successful. Considering first the Primary Objectives, a great deal of variability could be detected in the distribution of answers: nearly two out of three of the participants referring to Primary Objectives mentioned the action-taking/solution-finding dimension. The second biggest category was awareness-raising. This paints a picture that the participants mostly perceive messages about *Adaptive Action* and *Collaboration*, while few expressed comments about *Psychological Resilience* and *Organisational Relations*.

All Primary Objectives were found to be positively aligned with the original design intention, indicating success in conveying the original message to the audience. *Collaboration*, *Organisational Relations* and *Climate Science* were particularly successful in this regard. *Adaptive Action* largely aligns, yet a small number of participants expressed diverging experiences: some perceived that climate action was poorly elaborated, it was shallow, overly complex, not realistic, or easy to fail at. In

terms of *Psychological Resilience*, there was only one participant who perceived that the game added to their despair. However, given the infrequent mention of the category it ranks lowest in the alignment.

**Table 5. The ranking of design intentions within the Primary Objectives by frequency (% of both Primary and Secondary responses) and alignment with the original intent (-1 - +1 scale).**

| Ranking by frequency | Ranking by alignment |
|---|---|
| 1. Adaptive Action (21%) <br> 2. Collaboration (14%) <br> 3. Climate Science (12%) <br> 4. Organisational Relations (5%) <br> 5. Psychological Resilience (1.5%) | 1. Collaboration (1) <br> 2. Organisational Relations (0.94) <br> 3. Climate Science (0.9) <br> 4. Adaptive Action (0.8) <br> 5. Psychological Resilience (0.5) |

Of the Secondary Objectives, *Setting*, *Moderation* and *Relatability* were the most referenced, with *Setting* and *Relatability* positively aligning with the design intention. It should be noted that when controlling for an online versus in-person game experience, *Setting* shows the starkest contrast: the perception of the in-person experience is very positive, whereas the online one is noticeably lower, albeit still positively aligned. This contrast can be explained by the frequently cited technical difficulties reported by the online participants. *Team-building* ranked the highest in alignment with an overwhelmingly positive reception, but it was also one of the least mentioned design intentions.

*Moderation* and *Language* were the only two intentions that were negatively aligned with the original intention, with *Moderation* being the least aligned. While some participants reported enjoying the degree of facilitation, a large number of participants would have either liked to receive more, or conversely, less instructed gameplay. The *Language* intention was also negatively aligned and is closely related to *Moderation*. Participants experienced confusion in terms of game components and the instructions they were given, and some felt that trying to understand the game detracted from their capacity to reflect on the topic. However, other participants reported that the game was simple to understand.

**Table 6. The ranking of design intentions within the Secondary Objectives by frequency (% of both Primary and Secondary responses) and alignment with the original intent (-1 - +1 scale).**

| Ranking by frequency | Ranking by alignment |
|---|---|
| 1. Setting (15%) <br> 2. Moderation (14%) <br> 3. Relatability (7%) <br> 4. Language (6%) <br> 5. Team-building (4%) | 1. Team-building (1) <br> 2. Relatability (0.8) <br> 3. Setting (0.4) <br> 4. Language (-0.33) <br> 5. Moderation (-0.38) |

**5.2 Lessons learnt**

The purpose here was to advance the climate games and policy field by drafting guidelines for communicating adaptation to the public. Adaptation at a local level, among groups of non-professionals who are reliant on local trade-offs and knowledge exchange (Moser and Pike, 2015), can be facilitated via games, which create space for unordinary, and potentially transformative, conversations. MoD makes an interesting case study because of its focus on collective action and direct impact, as well as affective and relational learning, which are features seldom represented in climate game research    . Many games tend to focus on cognitive learning (Gerber et al., 2021) and take the underlying assumption that increasing knowledge on adaptation will lead to more adaptation. However, research demonstrates that it is not solely the lack of information forming a barrier to action (Fox et al., 2020; Panenko et al., 2021). Therefore, only focusing on measuring the degree of learning from a baseline to post-game may mislead one to think that barriers to action are being brought down.

This study diverges from such approaches by looking at the challenge from a different angle: how the intended messages are being perceived, and if the participants are being engaged in a way that appeals to them. Given that such a focus has not, to the knowledge of the authors, been tested previously, this paper adopted a qualitative approach to gain insights on what can be learnt by asking such questions. This section of the paper discusses the key findings and insights from the analysis.

**5.2.1 Inclusion of the participant perspective**

There is a tendency in communication research to treat participants as recipients of information instead of persons actively engaging in a dialogue with the communicators, giving meaning to climate change and action (Illingworth and Jack, 2018; Kumpu, 2022). There is a risk that in such cases only aspects that the communicator deems important are measured, which may result in omitting important participant perspectives. Given the concern that misunderstanding central game assumptions leads to iterations that do not bring about learning (de Kraker et al., 2021), deepening the understanding of the interaction between designers and participants is important. Intuitively, the importance grows when communication is targeted at audiences whose world view and learning methods significantly differ from that of the game designers: as is allegedly the case when climate professionals communicate adaptation to the public via games (Illingworth, 2020).

By focusing on this interaction, instead of learning, the method applied here helped discern both strong and weak aspects of the communication and served as the beginning of a conversation between designers, facilitators, and the target audience of the game. This, in turn, feeds into the monitoring and evaluation of the Game Day experience. Overall, the perception of the game is positive and aligns with the design intentions, which is an encouraging signal to develop similar designs or iterations of this game approach for similar non-professional audiences. As one participant summarised "This is definitely a very easy but effective way to engage my colleagues and friends about a serious subject of climate action", meaning that the game can help develop context and common language around the difficult topic.

Similar to other studies, the method used confirms that not only do individual game sessions lead to dissimilar results (Illingworth and Wake, 2021), but also that each audience member has unique perceptions of the messages conveyed.

Aggregating these results helps construct a picture of aspects that were most favourably regarded (approaching adaptive action

from collective and community/organisation level) and where the most distortion in communication emerged (engagement

strategy built around limited moderation and language used in the game).

**5.2.2 Collective action – communities and organisations at the system level**

Few adaptation measures are taken by single individuals, instead requiring collaboration on shared problems and negotiating

differences in opinions (Rumore et al., 2016). Nevertheless, the community or organisation-centred system level remains

mostly unexplored by climate games (Gerber et al., 2021). Much like other adaptation games, MoD conveys messages with

individualistic frames, breaking down complex scientific information to participants and pursuing cognitive learning, but it

also aims to achieve relational learning by addressing the collective (Flood et al., 2018). From a theory perspective, this could

create an out-of-the-ordinary scenario for the participants, which invites them to collectively explore alternative models for

action (Illingworth, 2020). Here, *Collaboration* and *Team-building* turned out to be most well-received by the participants,

signalling that this approach is welcomed as a way of communicating adaptive action to the general public. Participants shared

their key learning insights such as, "Collaboration must be done not only in the game but also in real life, because it would

help battle climate change and mitigate the pollutants and environmental pressures" and "Many people have interesting ideas

on what we can do. We should use more [of] the knowledge of the people around us and make it actionable"; and "Our actions

generate externalities and affect the most vulnerable groups. To achieve climate justice it is necessary to work as a team." This

shows clear support for the model adopted by the designers: a tactical decision simulation which requires collaborative

adaptation, and a narrative built around climate disruptions and team resilience.

Research has found that climate games sometimes struggle being relatable and relevant (Fjællingsdal and Klöckner, 2020),

however MoD succeeds in its intention of *Relatability*. This is encouraging given that if the audience perceives information as

relevant and engages with it in a dialogue, further action becomes more likely (Galeote et al., 2021). The reason for its

effectiveness here might have to do with the system level introduced: connecting knowledge, represented by *Organisational*

*Relations*, through the workplace guarantees a degree of familiarity and affection. Moreover, a good narrative is key for

decreasing abstraction for the public (Ouariachi et al., 2017) and relating the game to participants' experiences (Illingworth,

2020). The narrative of MoD presents a three-fold challenge common to most organisations: lack of time, resources, and

control. By playing together not all challenges are solved, but general resilience is gained, which appears to be a good pathway

on making climate change relatable for the general public.

Roleplay is frequently cited as an important factor contributing to learning through games (Parker et al., 2016; Flood et al.,

2018; Galeote et al., 2021; Fjællingsdal and Klöckner, 2020; Gerber et al., 2021). This case study confirms this in the sense

that immersing oneself into a game as a community member or a member of an organisation appears to be an effective way of

accessing the narrative. Additionally, this shows potential in triggering spill-over behaviour models from games to real life, as

the imagined threshold for action lowers (Ouariachi et al., 2017; Parker et al., 2016; Illingworth, 2020; Flood et al., 2018; Den

Haan and Van der Voort, 2018; Fjællingsdal and Klöckner, 2020). However, MoD also gives the option to roleplay different

characters - for instance, people in more vulnerable or powerful positions - which could contribute to relational learning as described by den Haan et al. (Den Haan and Van der Voort, 2018). This message was not referenced by any participant, however, showing preference for playing as oneself. This is not surprising given that the experience for participants unfamiliar with games or climate change can already be overwhelming by itself. It is suggested that this type of roleplay is possible and could lead to interesting reflections relevant for relational learning, though it is more likely achieved if the game experience was repeated a second time with the same group.

### 5.2.3 Online or in-person engagement?

Many climate games have the tendency to focus on digital rather than analogue experiences (Illingworth and Wake, 2019) and computers are often used to interact with the general public. While MoD should not be compared to virtual games as such, the case study did bring about interesting results when the answers were controlled for different game environments. The general experience was somewhat different as *Setting* and *Psychological Resilience* came out as much more prominent in the in-person setting compared with the online environment. This suggests that creating a fun and welcoming space, and addressing topics that require significant self-reflection might be more easily done in-person. At the same time, however, no evidence was found that communication was hindered in the digitised version, as found by other studies (Boomsma et al., 2018; Ho et al., 2022). For instance, the perception of *Collaboration* and *Team-building* did not suffer, though they were much less frequently mentioned. Nevertheless, the results suggest that the communicators should expect the experience to be somewhat different depending on the platform that is used and that if certain topics, in this case *Psychological Resilience*, are to be introduced, an analogue rather than digital space would be preferable.

### 5.2.4 Moderation

The designers and facilitators of MoD viewed having limited facilitation as a way to encourage participants to have a positive experience with experiential learning. In game research there are cases being made for those with high levels of moderation (Neset et al., 2020; Marome et al., 2021), autonomous gameplay with a non-obtrusive moderator (Ho et al., 2022; Tsai et al., 2021) as well as games where participants construct either the entire game, or parts of it, themselves (Lankford and Craven, 2020). MoD adopts a largely hands-off approach during the actual gameplay, focusing the facilitation on initial framing and debriefing the experience post-game, and prioritising autonomous gameplay during the session. This proved to be a controversial technique, with some participants praising it and others feeling frustrated and confused.

The participants would have liked to have seen both more and less moderation. For instance, one participant explains: "I liked the energy of the person introducing the game. Then when playing the game leaders did not really explain or introduce the game. They played along and answered questions. After a short while I felt a bit silly saying 'I don't understand'". Those who wanted more moderation implied that they were confused by the task at hand, which confirms that experiential learning of

games does not work in all contexts and can be itself a form of jargon (Illingworth, 2020). This highlights the need to strike a balance, especially with individuals with little experience with games, and explaining the purpose of experiential learning to them prior to the gameplay to reduce the confusion emerging around misaligned expectations.

At the same time, some participants experienced moderation very differently, for instance, according to one participant "It is great that the participants are trusted with the process, and that there is not too much intervention." Those who wanted less moderation, however, felt that the game rules, and especially the externally asserted time pressure, detracted from the quality of their discussions and degree to which they related to the game. This shows an interesting conflict between design intentions, as the time pressure is an important component of creating the game challenge, and generally appreciated by the participants. While discussion is an element mentioned by the designers (both in-game discussion and debrief) its importance in contrast with other design elements may have been underestimated. This is a quality uncovered by this study, which ought to be explored and tested in the next iteration of this game. As discussion is found to be the key to most of the learning in game communication (Neset et al., 2020), it seems that simply more time should be allocated; which is in line with the argument that the simpler and more familiar the game, the better participants are able to have simultaneous discussions and gameplay (Illingworth and Wake, 2021).

**5.2.5 General public as the target audience**

This study refers to the general public as an assortment of highly diverse groups. Their need for information, its reception, and trust toward it is bound to differ (Illingworth and Jack, 2018), and their experiences are difficult to homogenise. The *Climate Science* design intention, which was meant to capture the complexity of climate change, awareness, and urgency aligned strongly in both the online and in-person events. Theoretically, this intention would be closely tied to the *Language* design intention, as accessible language is a key component in expressing the complexity of the topic, yet this design intention was negatively aligned. This might indicate that those who did understand the decomplexified message reported it in the survey and, thus were categorised under *Climate Science* whereas those who struggled to follow referred to *Language*. As one participant reports: "It felt like I was the only outsider and all the others already knew some aspects of the game. There was a lot of jargon."

Games arguably have the potential to translate scientific knowledge making it accessible for the public (Gerber et al., 2021). However, designing the right amount of complexity into a game and finding optimal language is challenging as participants should not lose interest, but also not feel overwhelmed (Parker et al., 2016; Flood et al., 2018; Neset et al., 2020). This seems to be amplified when designing for the public whose experience with games and levels of knowledge are bound to vary. The role of facilitators is important with this audience type; moderation, and particularly its role during debrief, can unpack and explain jargon and tease out connections to real life (Neset et al., 2020). However, even if the discussion design element was connected to almost all design intentions of MoD, challenges emerged. This could suggest either that moderation/discussion is not performed in a way which would address everyone's needs, or, as previously found (Flood et al., 2018), that addressing

all needs within a short time window might simply be impossible and a series of engagements are needed. To resolve this issue, Neset et al. (Neset et al., 2020) propose that the same game could incorporate different levels of complexity which could be adjusted when needed.

Regardless, given that the overall reception was positive, this study reinforces the idea that games have a unique ability to cater to different needs, and this opens the conversation up to how games such as MoD can have increased relevance in the decision-making sphere. Games' ability to engage with diversity, be it in regard to attitudes, perception, behaviour, or cultural values, is what seems to make them so effective (Flood et al., 2018), and this presents a promising connection to using games as a way to help communities in, for example, local adaptation planning. Immersive experiences are needed to change the way that people relate to climate change (Bekoum Essokolo and Robinot, 2022), and it is encouraging to see that the general public shows eagerness to engage. The method applied here showcases clearly that when a game makes up such a complex package of information and is created to address different cognitive styles by including both textual, audiovisual and kinaesthetic aspects (Flood et al., 2018; Illingworth and Wake, 2021), the audience picks up on different features more strongly. The fact that collaboration was so positively reflected is an encouraging sign and demonstrates that games are effective when they create a sense of belonging and purpose for the participants (Illingworth, 2020) facing a shared problem they need to jointly tackle (Den Haan and Van der Voort, 2018). This can be designed to mimic the real-life circumstances of a community, as evident by a MoD iteration: a local advocacy tool co-created with a rural community in Kenya (Day of Adaptation, 2022). As positive local narratives correlate with the likelihood of action (Den Haan and Van der Voort, 2018), adaptation games such as this could ultimately serve as important tools to aid decision-making when adapted for specific local circumstances.

**6 Conclusions**

This paper presented a new method to study the designer-participant interaction in adaptation games, which takes a divergent approach to papers that focus on learning, or other analytical frameworks such as psychological distancing theory. Climate change and adaptation are experienced unequally around the world and this paper focuses specifically on communication within communities and organisations where the soft limits to adaptation can be influenced, by reprioritising resources to climate action (O'Neill et al., 2022). From this standpoint, the following key insights were uncovered:

1. Collaboration and team-building can be strongly recommended as frames for climate adaptation for the general public, as across the dataset they were found to align very well with the way the designers of Day of Adaptation intended. The results show that for the audience in question the actual knowledge shared in the game was less commonly reported as the key aspect, in comparison with the feeling of belonging and experience of solving challenges collectively.

2. Sometimes a game design may incorporate elements, which stand in conflict with each other, meaning that not all the objectives it sets out to achieve are synchronous. In the case of MoD, time pressure is designed within the game to

create a metaphor for the climate emergency, yet several participants found that the sense of emergency distorted their ability to discuss and brainstorm with their colleagues. While both objectives are important, the facilitator may have to make compromises to achieve one or the other.

3. Measuring both the number of design objectives as well as their relative distribution is important, as it can help the designers identify the stronger and weaker elements of their communication approach. For instance, while MoD effectively communicates aspects such as complexity of the human-environment system, few participants related the game to an increase in their psychological resilience. If the designers were to incorporate this objective as well, they might have to revisit some of the fundamental design assumptions they drafted, including considering how the varied past experiences that participants bring into the game may lead to emergent or unanticipated outcomes.

The reason for implementing a new method comes from the attempt to avoid replicating expert-to-public communication structures, which only focus on the participants as an object of study instead of looking at the whole game experience as a dialogical event (Illingworth, 2020). Knowing if a knowledge-action gap has been bridged is difficult to measure because of the complexity of predicting behaviour, however, participants aligning positively on climate action and reporting feelings of empowerment is a good indication of receptiveness to the messages being conveyed. Developing iterations based on such feedback could further enhance the effect, as could further exploring action-knowledge game structure over knowledge-action layouts (Crookall and Thorngate, 2009).

This approach is recommended to game designers and evaluators who are interested in discovering which of the messages they aim to communicate are perceived as intended and where distortion takes place, and to simply expand upon the understanding of the needs of those with whom they communicate. While ideally the dialogue with participants is more immediate, this approach was found to be less resource-intensive, and still enabled co-creation, given that the inputs are used to inform future iterations. For instance, here *Collaboration* outshone *Psychological Resilience*, and while both are important messages to convey about adaptation, they might be difficult to fit within one single activity. Insights such as this can help with modifying future iterations of the adopted approach and afford an identity and voice to the recipients of the communication.

The method can be improved in some parts, which could inspire some further research activities. First, if more information were obtained from individual participants, it would be possible to test not only the strongest categories on an aggregate level, but also if a single participant perceives all the design intentions. As it stands, the design intentions were sometimes artificially split, and for instance, the difference between the *Team-building* and *Collaboration* design intentions may have been too nuanced for the realities of a complex three-hour activity. Having higher resolution data would provide deeper understanding of the relationships between the categories, the degree to which the communication experience is different between participants, and what its determinants are. Additionally, having more representative group level data from each event would allow comparison between game events, which could lead to studying, for instance, the influence of group size and composition to the reactions. While there are reasons to assume that the participating groups have diverse backgrounds, the fact that the sample is neither randomised nor representative leaves some questions unanswered. A future research direction that would

move forward with a post-game survey designed to draw group-level conclusions without obscuring the diverse backgrounds of participants could help answer questions such as how to design for diverse audiences, and which factors best predict positive alignment.

Moreover, while the focus group gave an idea about which design elements related to the intentions, very few participants referred to specific elements, which makes it difficult to say with certainty which specific aspects might have been hindering or facilitating success. This presents a limitation of the design of the survey, but also a    further inquiry; a potential comparison of different elements aiming to achieve a similar intention would still be needed to understand strengths and weaknesses of specific elements. Finally, the method used to measure participant experience was easily skewed by negative experiences, which was most evident by the frustration with technical difficulties. This is a common issue known to survey research as well, as there is a tendency to report frustration over a session where no challenges emerge. Given the small size of the dataset this could still be considered within the results, as the researchers could look at each entry individually to see what fell under each design intention. If the study were to be scaled-up, a more sophisticated survey could be implemented, which would ask for feedback for all design intentions and elements. Ideally the participant experience would be captured during the game events as well, as this would provide a more complete snapshot of the game experience, off of which future tools could be based. based.

## 7 Appendices

**Appendix A: Post-game survey questions**

The following questions were presented in the post-game survey offered to all participants and used by the researchers to form the basis of the participant perspective for this study. Only open-field questions were included in this study, which are included in bold below.

1. **Please write down the first three (3) words that come to mind when describing your Game Day Experience.**
2. How would you rate your Game Day experience? (scale: 0-5)
    a. Please clarify if "unsatisfactory" or "improvement needed was selected
3. **What are the new perspectives or deeper understanding on climate action that you have gained on this topic, if applicable?**
4. **What is your key take-home message from the Game Day?**
5. How would you rate the organisation of the event? E.g., orderliness, easy to follow, engaging, etc. (scale: 0-5)
    a. Additional thoughts on the event organisation?
6. How would you rate the facilitator's performance? E.g., they explained things clearly, listened well, were engaging, etc. (scale: 0-5)
    a. Additional thoughts to share with the facilitators?
7. I would recommend this event to friends and colleagues. (scale: 0-5)

**8.**   Any other comments or suggestions?

9.   Age of participant

10.  Gender of participant

**Appendix B: Connections between design intentions and elements**

**Table B    1. A list of Design Elements Incorporated into the collaborative adaptation board game Minions of Disruptions. The**
**categorisation applies frameworks created by Gerber et al. [29], Lankford and Craven [30] and Razali et al. [35] to break down and**
**understand different game types and elements. Note that several design elements are connected to more than one design intention**
**and appear, therefore, several times in the table.**

| Design Intention | Design Element | Description |
|---|---|---|
| **Raising awareness:** *Climate Science* | **Aesthetic Experience** | Implicit messages are communicated via e.g. colours. For instance, the game board has carbon clouds which grow incrementally darker as emission levels increase and the climate impacts worsen. The purpose of this augmented sensory experience is to explain scientific concepts with the help of visuals and make memorization easier. |
| | **Audiovisual cues** | When the players hear the sound of a car engine they have to increase the difficulty level in the game. The purpose of this is to communicate urgency and draw a connection between the cause of climate change (emissions from driving) and the climate impacts. |
| | **Challenge: Time Constraints** | There is limited time to gain resilience; the feeling that time is running out creates a temporarily stressful ambiance and a sense of urgency. The purpose is to communicate the reality of the climate emergency. |
| | **Challenge: Uncontrollable Events** | There are aspects that players can control (i.e. actions), and that are out of their control (i.e. disruptions). This is a metaphor for climate change in the sense that some aspects of climate change can be locally influenced (i.e. adaptation), while addressing climate change as one organisation/community is impossible. |
| | **Discussion** | Players reflect on their experience and share local experiences and knowledge during and post-gameplay. The discussion is intended to empower questions and curiosity among players, but also to engage in the game by sharing their local knowledge about climate change. At the post-game discussion, the purpose is to create a space where the participants can pose open questions, and the game facilitators can further explain the mechanics of climate change. |
| | **Medium: Board** | The board models the structure of a community/organisation, mounting greenhouse gas emissions, and the climate impacts. The board limits the experience to a single shared reality, where climate change happens in real time (instead of in the distant future). |
| | **Medium: Cards for Action** | Action Cards inject information about possible mitigation and adaptation perspectives. From the point of view of climate science, the aim is to convey that climate change is anthropogenic, and thus, it is also possible to take action to prevent the worst impacts, if the action is timely. |
| | **Player Interaction: Collaboration / Competition between Teams** | The game is not limited to a single game board but there is a possibility to collaborate or compete between teams to share or mitigate emissions. The purpose of this element is to show the players the complexity of climate change, and the way that decisions taken locally have global spill-over effects. |

| Raising awareness: *Psychological Resilience* | Challenge: Time Constraints | There is limited time to gain resilience; the feeling that time is running out creates a temporarily stressful ambiance and a sense of urgency. The players are to perceive first-hand how decision-making may feel like when they have to respond to climate impacts/disasters on multiple fronts. |
|---|---|---|
| | Challenge: Uncontrollable Events | There are aspects that players can control (i.e. actions), and that are out of their control (i.e. disruptions). As the sense of limited power to influence can be taxing on individuals and communities, the game is intended to provide a safe space where this emotion can be explored. |
| | Discussion | Players reflect on their experience and share local experiences and knowledge during and post-gameplay. The possibility to share frustrations, joy and reflections with one's community is believed to be key in building trust and resilience. |
| Increase Knowledge, Understanding, Familiarity: *Organisational Relations* | Abstraction Level: Qualitative Description | A simplified model of the operations of a community/organisation and reality-check cards which connect local knowledge with abstract concepts (e.g. "what measures are in place in your community/organisation in case of a heatwave"). This element aims to increase knowledge about the players' organisations and the organisational readiness for climate change. |
| | Audiovisual Clues | When the players hear the sound of a car engine they have to increase the difficulty level in the game. This demonstrates a connection between organisational activity (e.g. company cars) and the causes of climate change. |
| | Challenge: Limited Funds | The amount of climate actions that a team can take is dependent on the funds they are in possession of; All teams start with the same amount of funding in the game, but their ability to gather funds depends on their strategic choices. This element conveys a common reality of most organisations, namely, that limited resources pushes the organisation to choose and prioritize between different actions. |
| | Challenge: Time Constraints | There is limited time to gain resilience; the feeling that time is running out creates a temporarily stressful ambiance and a sense of urgency. By introducing a stressful scenario in a game setting, the purpose is to foster connections between the individuals playing the game and train their ability to make decisions under pressure. |
| | Discussion | Players reflect on their experience and share local experiences and knowledge during and post-gameplay. The purpose of this element is to gather and share reflections about the current impact and perceived readiness of the organisation. |
| | Medium: Board | The board models the structure of a community/organisation, mounting greenhouse gas emissions, and the climate impacts. By showcasing the most essential functions of an organisation, the purpose of this element is to draw connections between functions and vulnerability. |
| | Player Interaction: Collaboration / Competition between Teams | The game is not limited to a single game board but there is a possibility to collaborate or compete between teams to share or mitigate emissions. This element is intended as a metaphor to explain how team collaboration can lead to more effective climate action, whereas dysfunctional team dynamics can hinder everyone's progress. |

| | | |
|---|---|---|
| | **Reward** | There are no lose-scenarios, and therefore all participants experience successful building of joint community/organisational resilience. |
| | **Role Play: Explicit Role Assignment with Optional Roleplay** | The participants play as equal members of a community or organisation, most commonly the one they take part in real life. If they so wish, they can also roleplay as a community/organisation that they do not belong in and/or assume characters and character powers which are inscribed by the game. Depending on which choice the team makes, the intention is to either deepen knowledge about one's own community/organisation, or a community/organisation of relevance. |
| | **Tactical Decision Simulation** | The players create a unique group strategy to inform their decision-making. Time, disruptions, limited funds and carbon accumulation are elements that make collaboration feel advantageous but also stressful. The players can experiment in a safe game setting how successful the team's collaboration is despite the stress it experiences. |
| **Promote Action-taking and Solution-finding:** *Adaptive Action* | **Challenge: Uncontrollable Events** | There are aspects that players can control (i.e. actions), and that are out of their control (i.e. disruptions). This is a metaphor for climate change in the sense that some aspects of climate change can be locally influenced (i.e. adaptation), even if addressing climate change as one organisation/community is impossible, and moreover, that the least beneficial thing is to do nothing. |
| | **Discussion** | Players reflect on their experience and share local experiences and knowledge during and post-gameplay. The discussion is intended to act as a catalyst for action and create a space for starting the discussion of how the given community/organisation could begin to take climate action. |
| | **Medium: Cards for Action** | Action Cards inject information about possible mitigation and adaptation perspectives. The purpose of these cards is to give real world examples of the array of possible actions, and also to convey that there are different scales at which action can be taken. |
| **Promote Action-taking and Solution-finding:** *Collaboration* | **Player Interaction: Collaboration/Competition between Teams** | The game is not limited to a single game board but there is a possibility to collaborate or compete between teams to share or mitigate emissions. If the teams collaborate, they are quicker to win the game, which is intended to signal that this is the case also in real life. |
| | **Player Interaction: Collaboration** | Although there are individual player turns, the player's team may help in decision-making. The aim here is to foster an experience that an individual does not need to face decision-making on their own, but that consultation and guidance from their community/organisation is beneficial and helpful. |
| | **Reward** | There are no lose-scenarios, and therefore all participants experience successful building of joint community/organisational resilience. |

| | | |
|---|---|---|
| | **Tactical Decision Simulation** | The players create a unique group strategy to inform their decision-making. Time, disruptions, limited funds and carbon accumulation are elements that make collaboration feel advantageous but also stressful. The players are guided to make collective decisions and create their very own team strategy out of several options. |
| **Achievable, Credible and Identity-driven:** *Relatability* | **Abstraction Level: Qualitative Description** | A simplified model of the operations of a community/organisation and reality-check cards which connect local knowledge with abstract concepts (e.g. "what measures are in place in your community/organisation in case of a heatwave"). The fact that local knowledge can be introduced to the game makes the game and climate change more relatable as the players can draw upon real life examples. |
| | **Audiovisual Cues** | When the players hear the sound of a car engine they have to increase the difficulty level in the game. Whilst there are many different causes to climate change, by choosing one that is close to the participants, and the emitting capacity of which is known by most, the mechanics of climate change become more evident. |
| | **Challenge: Uncontrollable Events** | There are aspects that players can control (i.e. actions), and that are out of their control (i.e. disruptions). Whilst playing as an omnipotent decision-maker might give a greater sense of influence, it is believed that the participants can better relate to a scenario where they are not able to prevent climate change from happening in the short time frame. |
| | **Discussion** | Players reflect on their experience and share local experiences and knowledge during and post-gameplay. In the discussion, the lived experience and the game experience can be connected. Moreover, an added purpose of the discussion is to create room for sharing experiences, feelings and self-reflections on climate change and action, which can enhance relatability. |
| | **Medium: Board** | The board models the structure of a community/organisation, mounting greenhouse gas emissions, and the climate impacts. On the game board, the players recognise familiar concepts and structures from their everyday life, which should help them form a connection between the game scenario and the player's actual life. |
| **Concrete:** *Language* | **Aesthetic Experience** | Implicit messages are communicated via e.g. colours. Using non-verbal language can be more memorable and easier to decode for some cognitive styles. |
| | **Kinaesthetic Experience** | The players move around cards, coins and pawns. The physical touch and concrete movements can be more memorable and easier to decode for some cognitive styles. |
| | **Character Design** | The basic climate action elements are presented as personified characters (Carbions, Climmies and Zillians, or carbon, climate disruptions, and climate action respectively). This adds an element of a story to the game, and aims to create more memorable images of concepts, which may be hard to memorize or understand. |
| | **Discussion** | Players reflect on their experience and share local experiences and knowledge during and post-gameplay. In the discussion any matters related to concepts that are unclear can be verbally elaborated. |

| | | |
|---|---|---|
| **Social and Reward-driven:** *Team-Building* | **Discussion** | Players reflect on their experience and share local experiences and knowledge during and post-gameplay. Sharing challenges, ideas and reflections can enhance team-building. |
| | **Moderation Type: Instructionist with constructionist elements** | The game rules are set and explained by facilitators, but the players are to learn the game experientially: no one controls for rule breaks. Players are given the possibility to inject their own knowledge into the game. Game organisers lead the post-discussion. The team will have to act autonomously during the game, fostering team-building. |
| | **Player Interaction: Collaboration/Competition between Teams** | The game is not limited to a single game board but there is a possibility to collaborate or compete between teams to share or mitigate emissions. This can foster team-building beyond the immediate team (game table) and more widely on the group level. |
| | **Player Interaction: Team Collaboration** | Although there are individual player turns, the player's team may help in decision-making. This cultivates a culture of supporting team members. |
| | **Reward** | There are no lose-scenarios, and therefore all participants experience successful building of joint community/organisational resilience. |
| | **Role Play: Explicit Role Assignment with optional role play** | The participants play as equal members of a community or organisation, most commonly the one they take part in real life. If they so wish, they can also roleplay as a community/organisation that they do not belong in and/or assume characters and character powers which are inscribed by the game. In either scenario (and especially in the role playing one) the team has to take into consideration different kinds of backgrounds, vulnerabilities and personalities. |
| | **Tactical Decision Simulation** | The players create a unique group strategy to inform their decision-making. Time, disruptions, limited funds and carbon accumulation are elements that make collaboration feel advantageous but also stressful. Collective strategy making can foster team-building. |
| **Fun, Meaningful and Reward-driven:** *Setting* | **Audiovisual Cues** | When the players hear the sound of a car engine they have to increase the difficulty level in the game. This sound may also add a layer of sensory experience and excitement. |
| | **Challenge: Time Constraints** | There is limited time to gain resilience; the feeling that time is running out creates a temporarily stressful ambiance and a sense of urgency. This also contributes to the game-like atmosphere, where players get engaged and motivated about the gameplay. |
| | **Discussion** | Players reflect on their experience and share local experiences and knowledge during and post-gameplay. This also creates the opportunity to create a safe space for learning and interaction. |

| | | |
|---|---|---|
| | **Moderation Type: Instructionist with constructionist elements** | The game rules are set and explained by facilitators, but the players are to learn the game experientially: no one controls for rule breaks. Players are given the possibility to inject their own knowledge into the game. Game organisers lead the post-discussion. Experiential learning is intended to give the players more room to engage. |
| | **Player Interaction: Collaboration/Competition between Teams** | The game is not limited to a single game board but there is a possibility to collaborate or compete between teams to share or mitigate emissions. This increases the dynamism of the game and creates the possibility for competitive interaction between teams. |
| | **Player Interaction: Team Collaboration** | Although there are individual player turns, the player's team may help in decision-making. This is intended to make the game more interactive. |
| | **Reward** | There are no lose-scenarios, and therefore all participants experience successful building of joint community/organisational resilience. |
| **Experiential learning:** *Moderation* | **Discussion** | Players reflect on their experience and share local experiences and knowledge during and post-gameplay. Discussion within the team is a key part in understanding the game rules and figuring out how the team will construct their game experience. In the meantime, the game organisers do help the players whenever they request for help or find themselves confused or lost. |
| | **Moderation Type: Instructionist with constructionist elements** | The game rules are set and explained by facilitators, but the players are to learn the game experientially: no one controls for rule breaks. Players are given the possibility to inject their own knowledge into the game. Game organisers lead the post-discussion. The constructionist elements are included to the game design to make sure that the players understand the game rules, and that they are correctly interpreting some themes, e.g. the mechanics of climate change. |
| | **Player Interaction: Team Collaboration** | Although there are individual player turns, the player's team may help in decision-making. The purpose of playing in a team is that no one is left behind and those that are slower to understand the game are able to follow thanks to the shared knowledge in the team. |

## Appendix C: The raw participant and statement numbers

**Table C1. Total number of participants and statements included in the analysis with breakdown between in-person and online events. Single Design Intention is the number of statements representing only one design intention. Two Design Intentions are the number of statements that were coded as having addressed multiple design intentions. Total Unique Statements represents the number of responses included for analysis; if a statement fit into two design intention categories, it was counted twice (Total Statements Analysed). Total Statements Omitted are those that would have required too much interpretation by the researchers.**

| | Total Participants | Single Design Intention | Two Design Intentions | Total Unique Statements | Total Statements Analysed | Total Statements Omitted |
|---|---|---|---|---|---|---|
| Question 2: How would you rate your Game Day experience? | | | | | | |
| Total | 140 | 20 | 4 | 24 | 28 | 7 |
| In person | 52 | 1 | 0 | 1 | 1 | 1 |
| Online | 88 | 19 | 4 | 23 | 27 | 6 |
| Question 3: What are the new perspectives or deeper understanding on climate action that you have gained on the topic, if applicable? | | | | | | |
| Total | 140 | 59 | 24 | 82 | 106 | 21 |
| In person | 52 | 15 | 8 | 23 | 31 | 8 |
| Online | 88 | 44 | 15 | 59 | 74 | 13 |
| Question 4: What is your key take-home message from the Game Day? | | | | | | |
| Total | 140 | 57 | 27 | 84 | 111 | 32 |
| In person | 52 | 18 | 7 | 25 | 32 | 12 |
| Online | 88 | 39 | 20 | 59 | 79 | 20 |
| Question 5: How would you rate the organisation of the event? E.g. orderliness, easy to follow, engaging, etc. | | | | | | |
| Total | 140 | 35 | 3 | 38 | 41 | 9 |
| In person | 52 | 11 | 0 | 11 | 11 | 4 |
| Online | 88 | 24 | 3 | 27 | 30 | 5 |
| Question 6: How would you rate the facilitator's performance? E.g. they explained things clearly, listened well, were engaging, etc. | | | | | | |
| Total | 140 | 15 | 6 | 21 | 27 | 25 |

| | | | | | |
|---|---|---|---|---|---|
| In person | 52 | 4 | 1 | 5 | 6 | 4 |
| Online | 88 | 11 | 5 | 16 | 21 | 21 |
| Question 8: Any other comments or suggestions? | | | | | | |
| Total | 140 | 11 | 5 | 16 | 21 | 21 |
| In person | 52 | 2 | 3 | 5 | 8 | 3 |
| Online | 88 | 9 | 2 | 11 | 13 | 18 |

**Data Availability.** This paper makes use of third-party data collected by Day of Adaptation for monitoring and evaluation purposes. Restrictions apply to the availability of these data. Data was obtained from Day of Adaptation and are available from the authors with the permission of Day of Adaptation.

**Author Contributions**. Conceptualisation, M.Ha., M.H., S.I., and M.S.; methodology, A.M., and M.S.; validation, M.Ha., and S.I.; formal analysis, M.S., and A.M.; investigation, M.H., A.M., and M.S.; data curation M.S.; writing— original draft preparation, M.H, and M.S.; writing—review and editing, M.Ha., M.H.,  S.I., A.M., and M.S. All authors have read and agreed to the published version of the manuscript.

**Conflicts of Interest.** Authors M.H, A.M. and M.S. have been involved as consultants at the non-profit Day of Adaptation. The sponsors had no role in the design, execution, interpretation, or writing of the study. S.I. is a member of the executive committee of journal Geoscience Communication. The peer-review process was guided by an independent editor.

**Ethical Statement.** This study was carried out according to the British Educational Research Association's (BERA) ethical guidelines for educational research, with all of the data in this study fully anonymised.

**Disclaimer.** All participant featured in the pictures in this paper have given explicit consent for distribution in media. This consent has been collected and is managed by Day of Adaptation.

**Acknowledgements.** The outline for this article was developed in parallel with its pair 'Decreasing Psychological Distance to Climate Adaptation through serious gaming: Minions of Disruptions' (published in Climate Services in December 2023), as two separate research questions emerged from the data gathered from Day of Adaptation's monitoring and evaluation effort. One of these questions related to the tangible impact of the game, which is assessed through the lens of psychological distancing in the aforementioned journal article. The other question, which deserved a reflection of its own, is the theme of this paper, namely how could such tangible impact be achieved by communicators, and which elements of multifaceted game-based communication would most readily be received by the public. To give space for the investigation of both research questions, two separate research teams were set up with the purpose to allow for broader reflections and make space for diversity of knowledge. While one of the datasets used in these two separate studies is largely looking at the same body of participants, the methods and angle through which the dataset is inspected diverge significantly.

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
