# Peer review of "A Collaborative Adaptation Game for Promoting Climate Action"

_EGUsphere, 2024_

## Community Comment (CC1)

**Comments on egusphere-2024-46**, by David Crookall, Apr 2024,

Minions of Disruptions™: A Collaborative Adaptation Game for Promoting Climate Action

My apologies for this rather short, and maybe curt, set of comments; I am getting ready to go to EGU24, and wish to send these before I leave.

| | |
|---|---|
| 1 | Title: Maybe consider swapping your title and subtitle. The main and immediately meaningful info should, in my view, come first. |
| 430 & 560 | "This study diverges …" and Concl. I would very much have liked to have had (more of) this in the introduction. Maybe I was misreading the introduction, but it is only when I got to the end that I suddenly realized (more) clearly what your method was. |
| 54 | "two separate datasets to form a dialogue between the designers' intentions and the audience's perception." The word dialogue bothers me. Could you not use 'comparison' or some other more literal term? It may be confused with dialogic teaching methods. |
| 129 | Consider using the abbreviation MoD for the game. |
| 60 | Consider listing the sections in a numbered list. |
| 67 | Games "function as communication vessels that transmit messages". Does this not hark back to the information-deficit model (p.35)? Simulation/games (in my view) are far from being vessels; they are player-co-constructed experiences in which relations and meanings are generated, sometimes quite irrespective of designer-intended messages. Hence the crucial need for debriefing. |
| 75 | For the knowledge-action gap in simulation/gaming, you might be interested in an article that I co-wrote years ago: https://doi.org/10.1177/1046878108330364 |
| 147 | Consider placing this table in an appendix. |
| 150 | It would be marvellous to have some photos of groups playing the game. |
| 164 | "Occasionally they are invited to share real-life knowledge and experiences,". One thing that many gamers tend to forget is that much/most of what happens in a game, and thus the game experience itself, depends largely on what players themselves bring to the game. Game designers tend to think that it is their design that determines everything in a game. No game would work if players suddenly left all their knowledge and skill behind as they enter the game. |

| | |
|---|---|
| 164 | Winning. Is MoD then a zero-sum game? If so, what competition arises, and how does this affect outcomes and the messages that people tale away from the game? |
| 167 | Debrief. This is crucial for any game. It is one of the main, if not the main, key to learning. It would be most useful to readers to provide more on how the debrief was conducted, what materials were used, the ways in which it helped people learn, what participants thought of the debriefing, etc. For more on debriefing, see my chapter https://www.researchgate.net/publication/374344073_Debriefing_A_Practical_Guide |
| 177 | Para. You start the para by outlining the drawbacks of the "most common way to evaluate games". Then you describe your method. Consider inverting the order: first describing your method, then saying how it overcomes common ways. The main point is not so much that your method overcomes drawbacks from other ways (although that is important), it is that your method is able to achieve the analysis and results that you want from your research objectives. This is where, I would have liked to have more about your method – maybe a para or two. |
| | This would be most useful to the gaming community as a whole, not just for geo-gaming. Explaining your method in more detail would help other gamers to do their evaluation. |
| | A general comment. The evaluation of a particular game is fraught with conundrums. Evaluating a game from a series of game plays is making a leap that should really be justified. We gamers are all convinced that a game works, is valid, is stupendous, etc, based on our experience with running games in general and with a particular run of a game. However, so much depends on the facilitation process and on the debriefing. Little wonder that the gaming community (outside geo-gaming) spends much effort on this. We are, in a word, making a leap of faith by inferring a game's value from our and participants' perception of the play. This raises the question (often debated in gaming circles) of the distinction between a game (the inert materials) and a play (the game brought life by people playing. We tend to speak as if the two are the same. They are manifestly totally different. This is reflected in many facilitators' experience of running the same 'game' (materials) and witnessing very varied play sessions (some highly successful, some a failure). Strictly speaking, instead of saying "this is a good game", we should say "this set of game materials often allow for good game-play experiences". This is partly why your new method of evaluation has great potential and should be explained in more detail. |
| 185 | "the gaming experience is received by participants". "received" bothers me. Would it not be more in keeping with a dialogic and/or constructive approach to use a word like |

| | 'seen' or 'experienced' or 'perceived' or 'lived'? |
|---|---|
| 185 | "future quantitative studies could be built". Most intriguing; it would be nice if you could elaborate on what is behind the "could be". |
| 229 | It would be marvellous if you could place a complete copy of the full "standardised post-game survey that all game participants were asked to fill out". After all, it seems to be a key element on your research findings, and it is also common for researchers to place these types of materials in an appendix. They are as important as the data that they generate. They would also allow a replication study, a standard approach in science. |
| 270 | Were these design intentions developed before the game was finalised or once the game had been used several times. I have a slight worry that knowledge of game play results might colour the expression of their intentions. |
| 300 | I must admit that I do not understand table 2. What are themes and statements? Also, the data in the table seem to be raw. How do the questions relate to the design intentions? Should the table not be in an appendix? |
| 306 | Fig 2. What is the purpose of the graph? My initial inclination was to think that it was to compare in person with online. How do you know the numbers behind the columns? For example, what makes it possible to assign a response (to what question) to an intention? |
| 312 | Fig 3. I have similar queries for this, especially about assigning responses to intentions. |
| 334 | What do the numbers in the text and in Table3 actually signify? For example, "Climate Science (0.900)"; is this a probability? My apologies, I must be missing something. I think that I would like more explanation of how you obtain these numbers and what they actually indicate. |
| 345 | In Fig 4, what do 1 and -1 represent? |
| 360 | Table 4. It seems that many elements contributed to each primary objective. Is there any way to highlight which elements contributed the most? Did you manage to 'measure' or get a sense of the holistic or overarching sense that players had of the game (or rather game plays) as a whole. Simulation/games are often said to provide players with a holistic sense of things – a gestalt is what one of our greatest gamers, Dick Duke, would say. See, eg, Duke, R. D. (1988). Gaming/simulation: A gestalt communication form. In D. Crookall & D. Saunders (Eds.), Communication and Simulation: From Two Fields to One Theme. Clevedon, Avon: Multilingual Matters. Also Duke, R. D. (2014). Gaming: The Future's Language. Second Printing. Bielefeld: |

Bertelsmann Verlag.  See also https://doi.org/10.1177/10468781231161955

| 368 | Table 5.  This looks most intriguing.  However, it would help me if you could (a) explain in a detailed example how you made the connections, and (b) what it says about how effective the runs of the game were. |
|---|---|
| 372 | What do you mean by "highly complex communication"? eg, as opposed to complex comm? |
| 422 | What field exactly do you wish to advance?  Environmental gaming? Climate simulation? Geo-games?  Gaming in general? |
| 426 | You assert that collective action etc feature seldom in other climate games.  My impression is that many actually do encourage these features, and in any case, it depends considerably on how the game is facilitated and debriefed. |
| 428 | You say that "research demonstrates".  It would be good to know what research – cite some examples. |
| 505 | This is one of the delicate aspects of facilitation.  How much it should be controlled by the facilitator and how much by the participants.  I see now easy answer, and each play will be different, depending on the needs of the participants (and also on the urges of the facilitator).  Some facilitators adopt a very hands-on approach no matter what; others a hands-off, or even no hands, in all circumstances.  (I discuss this in my debriefing chapter.) |

Overall, I very much liked the ms and the research method, even though I think, indeed am sure, that I did not manage to understand it fully.  I think that it will be a great asset to other gamers, in climate games or in general, wishing to assess the effectiveness of play sessions. The connection between play+debriefing sessions and the game materials themselves, with intervening variables such as facilitation style, data collection, etc. is in my view still fraught with problems, and is likely to be for quite some time to come.

Assuming that your ms is accepted for publication — I hope that it will be — I would very much like it to be included in the special issue of GC on the theme of climate and ocean education & communication – see https://oceansclimate.wixsite.com/oceansclimate/gc-special.  Let us see what the Editors say.

---

## Author Response (AR2)

| Reviewer | Reviewer comment | Response from the authors | Reference in the revised manuscript (with track changes) |
|---|---|---|---|
| Pratama Atmaja | The manuscript presents a novel contribution to the important yet emerging body of knowledge of games for complex issue communication. Due to their highly interactive and engaging nature, games are increasingly employed to communicate climate crises and other complex issues to the public. One glaring question regarding this communication method is how to ensure that the message is well-received, and this manuscript offers a preliminary yet compelling answer. It begins with providing a concise overview of the current literature on climate crises and games for climate-related communication and education. It then presents a case study of a specific board game for promoting climate actions. It then shows that the game's effectiveness can be measured by identifying the developer's design intentions, design elements, and the players' impressions of the intentions through a focus group and a survey. The manuscript discusses its research method, data, and results in great detail, culminating in design principles and directions for future research, the former useful for scholars and game developers wanting to replicate the game's success. | | |
| | At least that is what the manuscript seems to aim for. Unfortunately, its contributions are held back | In order to have this connection more elaborated, the revised manuscript has a considerably expanded | Row 740 onward. |

| | | somewhat by ambiguities. At the core of its case study, the manuscript presents three findings: (1) the developer's design intentions, (2) the players' impressions of the design intentions, and (3) some design elements that are supposed to transmit the intentions to the players. The first ambiguity revolves around the first and third findings. Looking at the design elements' descriptions, it is rather difficult to imagine exactly how some elements transmit some of the design intentions. For example, how exactly do "uncontrollable events" and "medium: board" transmit "relatability? The authors should elaborate on this more to make the findings more informative to other scholars and game developers. | Appendix 1, where each single design intention is connected with a design element followed by a brief description, which illustrates the connections, and provides the needed substance around the design elements. | |
| :--- | :--- | :--- | :--- | :--- |
| | | Additionally, the players' impressions of the developer's intentions are also riddled with ambiguity. For example, when some players complained about insufficient time for a discussion, why is this problem related to the "time constraints" element? Since "time constraints" typically apply to in-game activities, what does it have to do with the post-game discussion? Or was there actually an in-game discussion (which the development team did not mention somehow in the focus group)? Other than being confusing, such an ambiguity also indicates one thing: the design elements may not have actually encompassed every element of the game (understandably, the development team may have forgotten, or chosen not to mention, some actual elements in the focus group for some reason, including because they thought these elements were inessential to their design intentions). | The manuscript does describe discussion design elements relating to both debrief and in-game discussion. However, this was rather hidden in the Appendix, which is likely why the reviewer was confounded by the discussion in 5.2.4 Moderation. To clarify this specific part this paragraph was expanded and deepened. | Rows 605-607 |

| | | |
|---|---|---|
| | If this is the case, the authors should explicitly discuss these "unmentioned elements" to make the "transmission mechanism" of the design intentions clearer. | |
| | Regardless of these ambiguities, however, the manuscript's novelty and contributions remain worthwhile. Thus, we would recommend the manuscript's acceptance if the authors could resolve the issue. | |
| Johan Schaar | This is an important and novel contribution to research on the use of games to build awareness and insights on a highly complex issue. It expressly addresses the knowledge-action gap and barriers that stand in the way of acting on insights. It critiques the limitations of narrowly cognitive-focused knowledge-transfer approaches and seeks to understand designer-participant interaction with a critical focus on less studied affective and relational aspects that are often the keys to transformational learning. The approach is particularly important today given the troubling situation that neither mitigation nor adaptation action, at all levels, are keeping pace with the very tangible impacts of the rapidly changing climate. New approaches are needed to understand what prevents implementation of effective policies Games offer a promising tool both to gain a better understanding of impediments and find new ways forward. In this, the paper provides an account and learning from a promising test from which conclusions can be drawn for "guidelines for successful engagement. | |

| | | | |
|---|---|---|---|
| | The paper provides a very valuable, comprehensive and up-to-date review of the interdisciplinary research field | | |
| | **Specific comments** For the reader less familiar with the world of games, it would be useful if a brief but slightly more detailed description of how the game is provided could be given. | The 3.1.1 The gameplay section was expanded to provide a more detailed description of the game. | Rows 186-191 |
| | The participating groups are largely self-selected with no attempt to design randomized or representative groups. The consequences of this could be commented on. | This is now addressed in the conclusion where other limitations to the method are discussed. | Rows 696-700 |
| | The fact that participants represent a number of distinct and different institutions should allow some comparison between them in terms of perceptions, game outcomes and conclusions, both from a focus group and participant perspective. | The reason why this is not done is now further elaborated in the methods section, 3.2.2 The participant perspective | Rows 302-307 |
| | The 18 game events have taken place in a number of countries and in 4 continents. This indicates that the game is so general in character that it can successfully be introduced in very different contexts. A logical next step to increase its relevance as a tool that can create the foundations for real decision-making would be to adapt it to much more national/local circumstances. For example, it would be of much interest to see the game used in addressing close to real life and concrete trade-offs and tension of which there are many, as indicated in the description of the plan. It would be interesting to see the authors' views of how this kind | As mentioned, the game has proved to be successful in this and has been adapted to a rural community in Kenya. Whilst the space in the paper is much too limited for an in-depth discussion on this, a few elaborations were added under 5.2.5 General public as the target audience | Rows 632-647 |

| | | |
|---|---|---|
| | of application in specific governance settings and bio-physical and social contexts could be envisaged. Can it help decision-making under real-life uncertainty? | |
| | **Technical corrections**
There are few technical issues. The paper is long, probably prohibitive for many potentially interested readers. This could be remedied with a slightly more elaborate abstract. | The abstract has been re-written, with an attempt to be more elaborate and engaging. | Rows 10-34 |
| | Line 184: it says that the paper adopts a mixed-method approach. My understanding is that by mixed methods we usually mean the use of both qualitative and quantitative methods. But what is presented are different qualitative methods, not mixed methods. | This is a good correction, and we have removed the mention of mixed-methods. | Rows 239 |
| | 216. "A careful design of focus groups is key...". But the authors have not designed the focus group but have had to work with those that had actually acted as designers and facilitators. | Some of the authors were in fact present to design the focus group. We can see, however, how this section can be confounding, and therefore, we have written parts of this section | Rows 273-274 |
| David Crookall | My apologies for this rather short, and maybe curt, set of comments; I am getting ready to go to EGU24, and wish to send these before I leave. | | |
| | Title: Maybe consider swapping your title and subtitle. The main and immediately meaningful info should, in my view, come first. | The title and subtitle have been swapped. | Rows 1-2 |
| | "This study diverges ..." and Concl. I would very much have liked to have had (more of) this in the introduction. Maybe I was misreading the | A part of the introduction has been rewritten to more clearly describe the methods from the get-go and to clarify the part about the dialogue. | Rows 66-71 |

| | | |
|---|---|---|
| introduction, but it is only when I got to the end that I suddenly realized (more) clearly what your method was.

"two separate datasets to form a dialogue between the designers' intentions and the audience's perception." The word dialogue bothers me. Could you not use 'comparison' or some other more literal term? It may be confused with dialogic teaching methods. | | |
| Consider using the abbreviation MoD for the game. | Abbreviation MoD is now adopted throughout. | |
| Consider listing the sections in a numbered list. | We have added a numbered list. | Rows 78-86 |
| Games "function as communication vessels that transmit messages". Does this not hark back to the information-deficit model (p.35)? Simulation/games (in my view) are far from being vessels; they are player-co-constructed experiences in which relations and meanings are generated, sometimes quite irrespective of designer-intended messages. Hence the crucial need for debriefing. | This was not the intention, but we see how this part stands somewhat disconnected from the rest of the background. We have moved the first few sentences from the background, and attached them to discussion instead, with further elaboration. | Rows 458-461 |
| 75 For the knowledge-action gap in simulation/gaming, you might be interested in an article that I co-wrote years ago: https://doi.org/10.1177/1046878108330364 | Thank you for the material! We have visited this and thought it an excellent addition to the paper. | |
| 147 Consider placing this table in an appendix. | We have decided against moving this table in the appendix as we feel it contains important | |

| | | | |
|---|---|---|---|
| | | information for the reader to understand the context of participating groups. We have moved table 2 in the annex instead as it provides information of more supplementary character. | |
| | 150 It would be marvellous to have some photos of groups playing the game. | We checked with Day of Adaptation and this was indeed possible, so a few pictures have been added. | |
| | 164 "Occasionally they are invited to share real-life knowledge and experiences,". One thing that many gamers tend to forget is that much/most of what happens in a game, and thus the game experience itself, depends largely on what players themselves bring to the game. Game designers tend to think that it is their design that determines everything in a game. No game would work if players suddenly left all their knowledge and skill behind as they enter the game. | This is an excellent note and we wholeheartedly agree. | |
| | 164 Winning. Is MoD then a zero-sum game? If so, what competition arises, and how does this affect outcomes and the messages that people tale away from the game? | MoD is not a zero-sum game as all the participants who engage with the game will win, although some will be faster than others. However, what contributes to a competitive atmosphere is that the players are not necessarily aware of this as they start playing the game. From anecdotal experience, the competitive spirit contributes to excitement and motivation to take action more rapidly. This is unpacked in the Appendix 1. | |
| | 167 Debrief. This is crucial for any game. It is one of the main, if not the main, key to learning. It would be most useful to readers to provide more on how the debrief was conducted, what materials were used, the ways in which it helped people learn, what | We have expanded the section on debrief to help illuminate the aspects mentioned. Unfortunately, there was no specific mention of how debrief was experienced in the participant survey answers. | Rows 197 - 206 |

| | | |
|---|---|---|
| | participants thought of the debriefing, etc. For more on debriefing, see my chapter https://www.researchgate.net/publication/374344073 _Debriefing_A_Practical_Guide | |
| | 177 Para. You start the para by outlining the drawbacks of the "most common way to evaluate games". Then you describe your method. Consider inverting the order: first describing your method, then saying how it overcomes common ways. The main point is not so much that your method overcomes drawbacks from other ways (although that is important), it is that your method is able to achieve the analysis and results that you want from your research objectives. This is where, I would have liked to have more about your method – maybe a para or two.  This would be most useful to the gaming community as a whole, not just for geo-gaming. Explaining your method in more detail would help other gamers to do their evaluation.

A general comment. The evaluation of a particular game is fraught with conundrums. Evaluating a game from a series of game plays is making a leap that should really be justified. We gamers are all convinced that a game works, is valid, is stupendous, etc, based on our experience with running games in general and with a particular run of a game. However, so much depends on the facilitation process and on the debriefing. Little wonder that the gaming community (outside geo-gaming) spends much effort on this. We are, in a word, making a leap of faith by | We find this a very good suggestion and we have inverted the order, as well as expanded on the general method description. This will hopefully bring more substance to the method and clarify it. | Rows 213 - 239 |

| | | | |
|---|---|---|---|
| | inferring a game's value from our and participants' perception of the play. This raises the question (often debated in gaming circles) of the distinction between a game (the inert materials) and a play (the game brought life by people playing. We tend to speak as if the two are the same. They are manifestly totally different. This is reflected in many facilitators' experience of running the same 'game' (materials) and witnessing very varied play sessions (some highly successful, some a failure). Strictly speaking, instead of saying "this is a good game", we should say "this set of game materials often allow for good game-play experiences". This is partly why your new method of evaluation has great potential and should be explained in more detail. | | |
| | 185 "the gaming experience is received by participants". "received" bothers me. Would it not be more in keeping with a dialogic and/or constructive approach to use a word like 'seen' or 'experienced' or 'perceived' or 'lived'? | We fully agree with this comment and have changed the word 'received' into 'perceived' in most instances. | |
| | 185 "future quantitative studies could be built". Most intriguing; it would be nice if you could elaborate on what is behind the "could be". | We explain further details in 6. Conclusions | |
| | 229 It would be marvellous if you could place a complete copy of the full "standardised post-game survey that all game participants were asked to fill out". After all, it seems to be a key element on your research findings, and it is also common for researchers to place these types of materials in an appendix. They are as important as the data that | We had already detailed the set of questions in Table 2, but agree with disclosing the whole survey to improve replicability. This is now Appendix 2. | |

| | they generate. They would also allow a replication study, a standard approach in science. | | |
|---|---|---|---|
| | 270 Were these design intentions developed before the game was finalised or once the game had been used several times. I have a slight worry that knowledge of game play results might colour the expression of their intentions. | This is a valid concern, and given that the game was designed many years ago, we do not have access to original intent, but an intent that has undoubtedly been coloured by new ideas and thoughts among the designers and facilitators since the creation of the game. If one would like to see the very original intent, this ought to be researched at the very moment when the game is first created.

Nevertheless, we believe this bias would have been more worrisome, if all design intentions had been aligned with the participant perception. This could have suggested that the focus group participants had already skewed their perception based on participant feedback. To be sure, as a matter that wouldn't be a bad thing at all, however, it would of course limit the extent to which this method is able to create useful knowledge.

Given that in our results we see discongruence, we are confident in either that

a) The designers had not been influenced by knowledge of game play results

and/or

b) Despite some knowledge from game play results, the designers do not have a | |

| | | complete picture of the participant perception.

This, therefore, shows that the method has potential in increasing the knowledge of designers and facilitators. | |
|---|---|---|---|
| | 300 I must admit that I do not understand table 2. What are themes and statements? Also, the data in the table seem to be raw. How do the questions relate to the design intentions? Should the table not be in an appendix? | Yes, we agree that this table has mostly supplementary value, and it has now been moved to the appendix, with some added elaborations | Rows 740 - |
| | 306 Fig 2. What is the purpose of the graph? My initial inclination was to think that it was to compare in person with online. How do you know the numbers behind the columns? For example, what makes it possible to assign a response (to what question) to an Intention?

312 Fig 3. I have similar queries for this, especially about assigning responses to intentions. | The purpose of this graph is to demonstrate how many times a certain design intention was mentioned in the participant survey. It includes the comparison between the in-person and online to see which categories were most frequently mentioned in both. The numbers are a result of the analysis conducted, which is explained in the methods section (3.3. The analysis). This has been now further elaborated to crystallise how we arrive at the numbers. The assignment happened based on the results from the focus group, which are detailed in 4.1 The design intent | Rows 319-320

Rows 337-354 |
| | 334 What do the numbers in the text and in Table3 actually signify? For example, "Climate Science (0.900)"; is this a probability? My apologies, I must be missing something. I think that I would like more explanation of how you obtain these numbers and what they actually indicate. | Some elaboration was added to Table 3. Please also refer to the methods section (3.3. The analysis) | Rows 319-320

Rows 407-408 |

| | | | |
|---|---|---|---|
| | 345 In Fig 4, what do 1 and -1 represent? | | |
| | 360 Table 4. It seems that many elements contributed to each primary objective. Is there any way to highlight which elements contributed the most? | This was our initial wish, however, the mentions of specific elements were so scarce in the material we were using that it is difficult to get a grasp of what element might have contributed the most, thus, we sticked with the general picture.

See the discussion in the conclusion also. | Rows 701-705 |
| | Did you manage to 'measure' or get a sense of the holistic or overarching sense that players had of the game (or rather game plays) as a whole. Simulation/games are often said to provide players with a holistic sense of things – a gestalt is what one of our greatest gamers, Dick Duke, would say. See, eg, Duke, R. D. (1988). Gaming/simulation: A gestalt communication form. In D. Crookall & D. Saunders (Eds.), Communication and Simulation: From Two Fields to One Theme. Clevedon, Avon: Multilingual Matters. Also Duke, R. D. (2014). Gaming: The Future's Language. Second Printing. Bielefeld:

Bertelsmann Verlag. See also https://doi.org/10.1177/10468781231161955 | Thank you for the interesting reading material! For the purpose of our study and the measurements we did, the focus on the alignment is what we sought after the most, however, we are aware based on the other survey questions, not included as data into this study, that most players were content with the experience and would recommend it to others. | |
| | 368 Table 5. This looks most intriguing. However, it would help me if you could (a) explain in a detailed example how you made the connections, and (b) what it says about how effective the runs of the game were. | We arrived at the classification through the means of a qualitative review, in other words, reading into the Ouariachi et al. framework and contrasting this with the data collected from the focus group. Due | Rows 443-444 |

| | | to the length of the paper, we will not be adding much detail around this process, but an explanatory note has now been added to the Table description.

The framework was chosen as it is a widely recognised one and the authors have done a great deal of work to create a framework that would help understand different possible game mechanics and climate engagement, and was, therefore, considered an appropriate choice here. | |
|---|---|---|---|
| | 372 What do you mean by "highly complex communication"? eg, as opposed to complex comm? | We see that this was a somewhat unnecessary emphasis word, and have removed 'highly', as 'complex' is sufficient to describe our meaning. | Row 447 |
| | 422 What field exactly do you wish to advance? Environmental gaming? Climate simulation? Geo-games? Gaming in general? | We have specified this to be "climate games and policy field" | Row 501 |
| | 426 You assert that collective action etc feature seldom in other climate games. My impression is that many actually do encourage these features, and in any case, it depends considerably on how the game is facilitated and debriefed. | Thank you for this comment. The purpose here was to state that this is not often represented in research, although we too share your impression that there are other climate games with features encouraging collective action. We have adjusted the formulation to be congruent with what we argue in the Background section | Rows 505-506 |
| | 428 You say that "research demonstrates". It would be good to know what research – cite some examples. | This was an oversight, thank you for bringing it to our attention. The appropriate references have been added. | Rows 507-508 |
| | 505 This is one of the delicate aspects of facilitation. How much it should be controlled by the facilitator | | |

| | | |
|---|---|---|
| | and how much by the participants. I see now easy answer, and each play will be different, depending on the needs of the participants (and also on the urges of the facilitator). Some facilitators adopt a very hands-on approach no matter what; others a hands-off, or even no hands, in all circumstances. (I discuss this in my debriefing chapter.) | |
| | Overall, I very much liked the ms and the research method, even though I think, indeed am sure, that I did not manage to understand it fully. I think that it will be a great asset to other gamers, in climate games or in general, wishing to assess the effectiveness of play sessions. The connection between play+debriefing sessions and the game materials themselves, with intervening variables such as facilitation style, data collection, etc. is in my view still fraught with problems, and is likely to be for quite some time to come.

Assuming that your ms is accepted for publication — I hope that it will be — I would very much like it to be included in the special issue of GC on the theme of climate and ocean

education & communication – see https://oceansclimate.wixsite.com/oceansclimate/gc-special. Let us see what the Editors say. | |